Myers *et al. Genome Biology*　(2024) 25:130

## METHOD

# HATCHet2: clone- and haplotype-specific copy number inference from bulk tumor sequencing data

Matthew A. Myers[1], Brian J. Arnold[2], Vineet Bansal[3], Metin Balaban[1], Katelyn M. Mullen[5], Simone Zaccaria[4]* and Benjamin J. Raphael[1]*

*Correspondence:
s.zaccaria@ucl.ac.uk;
braphael@princeton.edu

[1] Department of Computer Science, Princeton University, Princeton, USA
[2] Center for Statistics and Machine Learning, Princeton University, Princeton, USA
[3] Princeton Research Computing, Princeton University, Princeton, NJ, USA
[4] Computational Cancer Genomics Research Group, University College London Cancer Institute, London, UK
[5] Human Oncology & Pathogenesis Program, Memorial Sloan Kettering Cancer Center, New York, NY, USA

## Abstract

Bulk DNA sequencing of multiple samples from the same tumor is becoming common, yet most methods to infer copy-number aberrations (CNAs) from this data analyze individual samples independently. We introduce HATCHet2, an algorithm to identify haplotype- and clone-specific CNAs simultaneously from multiple bulk samples. HATCHet2 extends the earlier HATCHet method by improving identification of focal CNAs and introducing a novel statistic, the minor haplotype B-allele frequency (mhBAF), that enables identification of mirrored-subclonal CNAs. We demonstrate HATCHet2's improved accuracy using simulations and a single-cell sequencing dataset. HATCHet2 analysis of 10 prostate cancer patients reveals previously unreported mirrored-subclonal CNAs affecting cancer genes.

**Keywords:** Copy-number aberrations, Cancer, Genomics, Tumor heterogeneity, Clone, Allele-specific, Haplotype, DNA sequencing

## Background

Somatic *copy-number aberrations* (CNAs) are frequent genetic alterations in cancer that increase or decrease the number of copies of relatively large genomic segments. CNAs range in size from focal events affecting hundreds to thousands of nucleotides, to gain and loss of chromosome arms or whole chromosomes, and even whole-genome duplication (WGD) in which the number of copies of every chromosome is doubled. CNAs are nearly ubiquitous in human cancers [1, 1–7] and drive tumor progression through amplification of oncogenes or inactivation of tumor suppressor genes [8]. CNAs are highly predictive of prognosis [9, 10] and can lead to resistance to treatment [11].

Nearly all cancer sequencing studies-particularly of large cohorts (e.g., The Cancer Genome Atlas (TCGA) [5] and Pan-cancer Analysis of Whole Genomes (PCAWG) [12]) as well as clinical sequencing of tumor specimens [13, 14]-analyze bulk tumor samples containing a mixture of thousands to millions of different normal and cancer cells that

are sequenced simultaneously. Moreover, the cancer cells themselves may contain different complements of CNAs, reflecting the distinct subpopulations of cancerous cells, or clones, present in a tumor sample. Thus, a bulk tumor sample produces a mixed signal from different clones with different CNAs. While single-cell DNA sequencing of tumors would obviate the complications of a mixed sample, single-cell technologies remain specialized techniques [15–19] and have not yet been applied to large cohorts of tumor samples.

Many computational methods have been developed to solve the challenging problem of identifying CNAs from bulk samples [20–38], but nearly all of these analyze individual bulk samples. Recently, several studies have sequenced multiple bulk tumor samples from each individual patient, enabling researchers to profile the heterogeneity of the disease in different tumor regions [39–41], across primary and metastatic sites [42–44], and over time [40, 45]. The CNAs and clones present in multiple samples are related by the shared evolutionary history of the tumor, but few methods leverage these relationships via joint analysis of multiple samples. HATCHet [30] is one algorithm that identifies CNAs simultaneously from multiple bulk samples, and it has two key advantages over previous methods. First, HATCHet identifies *clone-specific* CNAs across distinct tumor clones present in multiple samples, meaning that CNAs are assigned to a small number of distinct clones and co-occurrence of CNAs in the same clone is inferred. While previous methods have also aimed to identify subclonal CNAs present in different clones, most of these methods did not assign CNAs to distinct clones, which provides important information for evolutionary and phylogenetic analyses [46, 47], as well as additional constraints for improving the accuracy of CNA identification [30]. Second, HATCHet includes a model selection criterion to evaluate the trade-off between subclonal CNAs and the presence of WGDs, which in turn allowed the algorithm to overcome the error-prone inference of tumor ploidy. These features led to significant improvements in the accuracy of CNA identification from multi-region sequencing data [30].

Here, we introduce HATCHet2 (Holistic Allele-specific Tumor Copy-number Heterogeneity 2), a method to infer clone-specific and haplotype-specific CNAs from DNA sequencing data from one or more bulk tumor samples from the same patient (Fig. 1). HATCHet2 has two key advantages over previous methods, including HATCHet. First, HATCHet2 identifies *mirrored-subclonal CNAs*, which are events in which both of the two parental haplotypes are altered by CNAs in distinct tumor clones. We refer to such events as mirrored-subclonal CNAs because the more-abundant haplotype is opposite ("mirrored") between the two clones. Previous efforts to detect these events have been limited to the identification of *mirrored-subclonal allelic imbalance* [39, 48–50], a sample-level rather than clone-level feature, in which the more abundant haplotype on average in a sample is different between samples. This analysis treated samples as homogeneous and did not identify haplotype-specific copy-number states nor assign them to specific tumor clones. HATCHet2 introduces a statistic, the *minor-haplotype B-allele frequency*, which measures haplotype-specific fluctuations across samples and enables HATCHet2 to identify mirrored-subclonal CNAs and assign them to tumor clones.

Second, HATCHet2 improves the identification of clone-specific focal CNAs across multiple samples by introducing a new method for dividing the genome into *variable-width* bins as well as computing a *local-global* clustering with a hidden Markov

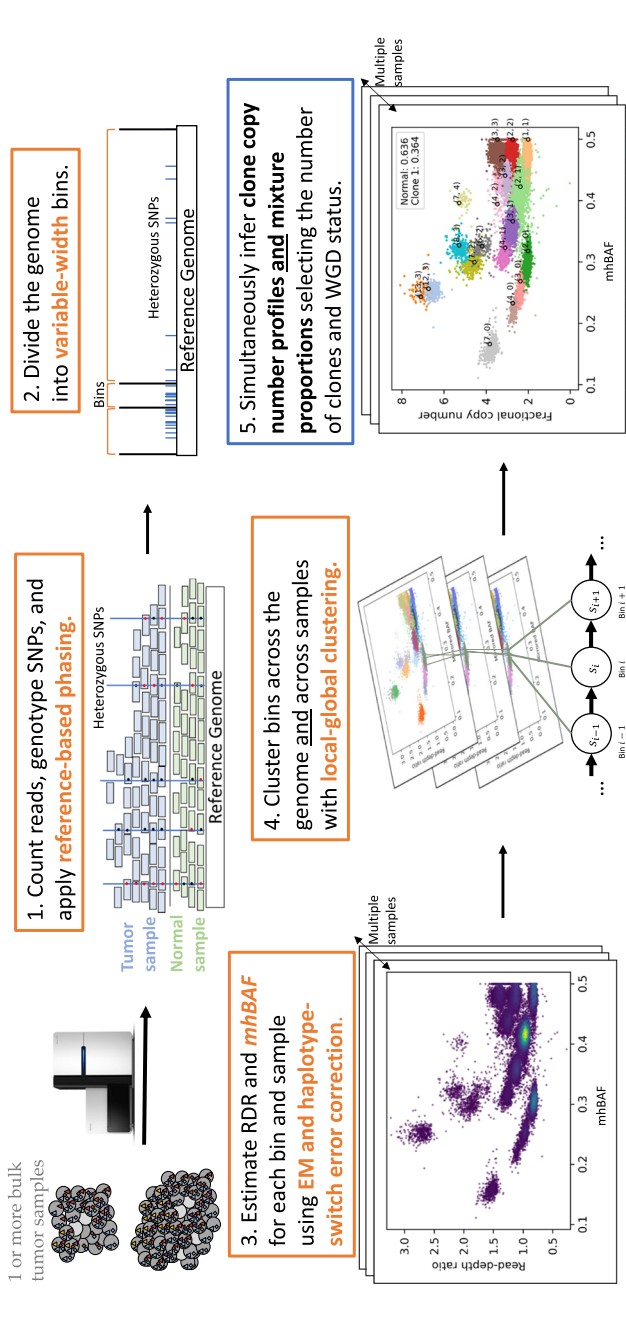

**Fig. 1** Overview of HATCHet2. Starting from aligned DNA sequence reads from one or more bulk tumor samples from the same patient, HATCHet2 identifies integer-valued copy-number profiles for multiple tumor clones along with the proportion of each clone in every sample. Steps in orange boxes indicate key methodological improvements in HATCHet2 compared to HATCHet [30], with orange text highlighting the new features in each step

model (HMM). This clustering algorithm combines advantages of local segmentation approaches that have proven useful for analysis of individual samples [28, 29, 34–36] with the global clustering algorithm introduced in HATCHet to identify CNAs from multiple sample simultaneously and assign them to clones. We show that HATCHet2 has higher power to detect focal CNAs that are missed by HATCHet, while also assigning these focal CNAs to distinct tumor clones across samples (which is impossible using methods that analyze each sample separately).

In addition to these methodological improvements, HATCHet2 includes numerous software features, such as automated phasing of germline variants, cloud-based execution, bioconda [51] installation, support for open-source ILP solvers, and continuous integration to automate testing. These features make HATCHet2 an easily automated tool that can be applied to large and cloud-based cohort datasets such as those generated by recent cancer sequencing studies.

We show that HATCHet2 outperforms HATCHet and several other copy-number inference methods on simulated multi-sample and single-sample datasets. We use HATCHet2 to analyze 10 prostate cancer patients with multi-sample WGS [42] and show that HATCHet2 recovers focal copy-number events more reliably than previous analyses using Battenberg [23] or HATCHet [30]. In this cohort, we identify mirrored-subclonal copy-number aberrations in all 10 of these cases, including parallel amplification of *ELK4* and *SLC45A3* which comprise a chimeric transcript known to be related to cell proliferation in prostate cancer [52, 53]. Additionally, we apply HATCHet2 to multi-region single-cell WGS data [54] (treating each region as a bulk sample) and find that HATCHet2 recovers the major tumor clones and mirrored-subclonal CNAs documented in a previous single-cell analysis of this dataset [15].

## Results

### HATCHet2 algorithm

HATCHet2 infers integer-valued haplotype-specific copy-number profiles for multiple distinct tumor clones from DNA sequencing data from one or more tumor samples from the same patient. HATCHet2 also infers the proportions of each clone in every sample. Unlike previous methods and similarly to HATCHet [30], HATCHet2 infers CNAs jointly from multiple samples while assigning each inferred CNA to a distinct clone. In comparison with HATCHet, HATCHet2 has two substantial improvements: first, the ability to infer *haplotype*-specific copy-number profiles and assign them to distinct clones, which enables the identification of clone-specific mirrored-subclonal CNAs; second, more accurate identification of focal copy-number aberrations. These improvements are enabled by several methodological innovations, which we summarize in the five main steps of HATCHet2. Additional details of each step are provided in the "Methods" section.

The first step of HATCHet2 identifies heterozygous germline single-nucleotide polymorphisms (SNPs) from the matched normal sample using a reference database (e.g., dbSNP [55]) and applies reference-based phasing [56, 56, 57] to group SNPs that likely belong on the same haplotype together into haplotype blocks. This step improves the estimation of allelic imbalance, especially from low-coverage or low-purity DNA sequencing data such as cell-free DNA [58–61] or low-pass WGS [62–65]. We apply a

statistical test in HATCHet2 to ensure that the SNPs in each haplotype block are likely to be on the same haplotype in the tumor genome. This step also counts the number of sequencing reads that cover each position in the genome, which are used in subsequent steps.

The second step of HATCHet2 partitions the reference genome into genomic bins of variable width (Fig. 1). Specifically, HATCHet2 partitions the reference genome into variable-width bins that each have an approximately equal number of reads covering heterozygous germline SNPs. Compared to fixed-width bins used in many approaches, HATCHet2's variable-width bins account for the varying density of germline SNPs across the genome to stabilize the variance of the allelic imbalance signal while guaranteeing sufficient power to obtain high-quality analysis of haplotype frequencies.

The third step of HATCHet2 computes the read-depth ratio (RDR) and a new statistic called the *minor haplotype B-allele frequency* (mhBAF) for each bin and each sample (Fig. 1). The RDR is proportional to the total number of copies of a genomic region in the sample, while the mhBAF quantifies the imbalance between haplotypes across clones. Specifically, the mhBAF measures the allelic imbalance of a genomic bin from germline heterozygous SNPs while accounting for variable gain/loss of the two parental copies of a locus across different samples. The mhBAF aggregates imbalance across all SNPs in a genomic bin, as opposed to the standard B-allele frequency (BAF) statistic used by many existing methods for CNA identification [25, 28, 31, 35], which is specific to each SNP position. Unlike other aggregate statistics such as the mirrored BAF (mBAF) [30] or the squared log-odds ratio [31], the mhBAF quantifies the frequency of a specific haplotype across samples, which HATCHet2 uses to identify haplotype-specific CNAs and assign them to tumor clones. To compute mhBAF in HATCHet2, we develop an expectation-maximization algorithm to compute a pseudo-maximum likelihood estimate for the mhBAF across different bulk tumor samples by extending existing approaches that have been introduced for single-cell analysis [15, 66].

The fourth step of HATCHet2 groups genomic bins into clusters jointly across all samples. The goal of this step is to identify segments of neighboring genomic bins that are affected by the same complement of CNAs across tumor clones. Existing methods perform this step either by segmenting the genome into intervals-leveraging *local* agreement between copy-number states at adjacent genomic locations in individual samples [35, 36, 67]-or by clustering bins across the genome [31, 32] and across multiple samples [30]-leveraging *global* genomic information. HATCHet2 uses a *local-global clustering* algorithm that combines features from these two previous approaches, improving the identification of focal CNAs while also leveraging global information along genome and across samples. Specifically, HATCHet2 uses a simplified hidden Markov model (HMM) in which global parameters are inferred along the genome and across samples, and CNAs are identified using local genomic signals. Although HMMs are used in multiple previous methods for identification of CNAs, HATCHet2's model aims simply to group bins into copy-number states, unlike existing approaches which aim to concurrently infer the underlying integer copy numbers [24, 29, 68–71].

The fifth and final step of HATCHet2 infers integer-valued haplotype-specific copy-number profiles for $K$ distinct tumor clones as well as the proportion of each clone present in each sample. Specifically, we model the haplotype-specific copy-number profile

of each clone $i$ as a pair $(\mathbf{a}_i, \mathbf{b}_i)$ of integer vectors, where each pair $(a_{i,s}, b_{i,s})$ represents the copy numbers of the two haplotypes for each genomic region $s$. We define the *clone proportion* $u_{i,p}$ to be the proportion of cells in sample $p$ that belong to clone $i$. HATCHet2 infers the copy numbers and clone proportions from the RDR and mhBAF of each cluster using the multi-sample matrix factorization approach from HATCHet, which derives $(\mathbf{a}_i, \mathbf{b}_i)$ while simultaneously inferring the number of tumor clones, their proportions in each sample, and the occurrence of WGDs.

HATCHet2 also includes several additional improvements in software infrastructure, as detailed further in the "Methods" section.

### Benchmarking on simulated data

We compared HATCHet2 to HATCHet [30], TITAN [24], cloneHD [25], and Battenberg [23] on 32 tumor samples from eight different datasets generated using the MASCoTE simulation framework [30]. We also evaluated performance on single-sample datasets by considering each sample independently. To evaluate performance on these datasets, we computed (1) accuracy, which evaluates whether the set of inferred states matches the ground truth states for each genomic segment (weighted by segment length), and (2) average allele-specific accuracy per genome position (AASAPGP), which additionally weights the accuracy by the clone proportions associated with each state and segment in each sample. Additional details on the simulated datasets and evaluation metrics are reported in Additional file 1: Section S1, and additional results focusing on simulated event sizes and clone proportions are reported in Additional file 1: Section S5.

Overall, we observe that HATCHet2 and HATCHet [30] outperform cloneHD [25], Battenberg [23], and TITAN [24] on these simulations, with better accuracy and AASAPGP: on average, HATCHet and HATCHet2 achieved 2.28 times higher accuracy and 1.50 times higher AASAPGP than the best non-HATCHet method, which was typically Battenberg or cloneHD (Additional file 1: Fig. S1A-B). HATCHet2 outperformed HATCHet in terms of AASAPGP on these multi-sample datasets (paired *t*-test $p = 0.046$) but not in terms of accuracy (paired *t*-test $p = 0.78$). While there was no significant difference between HATCHet2 and HATCHet on the multi-sample datasets, HATCHet2 outperformed HATCHet on single-sample datasets (median improvement in accuracy by 2.0% and AASAPGP by 1.4%; Additional file 1: Fig. S1C-D). This difference was driven mainly by performance on tetraploid datasets (median improvement in accuracy by 3.8% and AASAPGP by 2.0%). HATCHet2 also showed a small improvement on the diploid datasets (median improvement in accuracy of 1.3% and AASAPGP of 0.8%).

We show in the supplement on additional simulated datasets that HATCHet2 outperforms HATCHet in recovering tumor clones with varying purity (Additional file 1: Section S7) and in identifying mirrored-subclonal CNAs (Additional file 1: Section S9).

### Identification of clone-specific focal CNAs

We applied HATCHet2 to 50 bulk tumor samples from 10 prostate cancers analyzed by Gundem et al. [42]. We compared the copy-number profiles obtained by HATCHet2 to those obtained by Battenberg [23] in the original publication [42], and

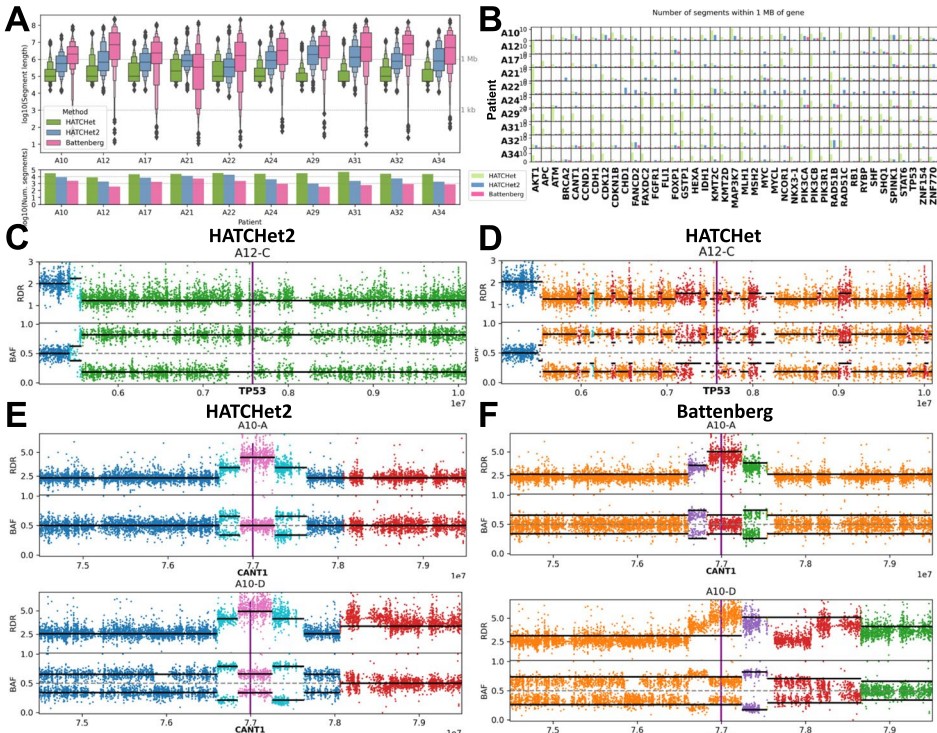

**Fig. 2** Copy-number segments identified by HATCHet2, HATCHet [30], and Battenberg [23] in prostate cancer. **A** (Top) Lengths of segments identified by the three methods from 10 prostate cancers from Gundem et al., 2015 [42]. Dotted gray lines indicate 1 kilobase and 1 megabase. (Bottom) Numbers of segments inferred by each method on each prostate cancer patient. **B** Number of segments identified by each method for each patient (row) within 1 megabase of 41 genes (columns) from The Cancer Genome Atlas prostate cancer publication [72]. **C** Copy-number segments identified by HATCHet2 near the *TP53* locus on chromosome 17 in one sample from patient A12. (Two other samples from this patient are shown in Additional file 1: Fig. S3.) Each point is a small genomic region that contains exactly one SNP with indicated read-depth ratio (RDR) and B-allele frequency (BAF) and is colored by the assigned copy-number state for the corresponding method. Black bars indicate the expected RDR and BAF of each segment according to the copy-number states and clone proportions assigned by the corresponding method. Gene location is indicated by vertical purple bar. Full copy-number state legends for panels **C**–**F** are reported in Additional file 1: Fig. S5. **D** Copy-number segments identified by HATCHet near TP53 for the same sample as panel **C** (Battenberg results for this patient are shown in Additional file 1: Fig. S3). **E** Copy-number segments identified by HATCHet2 and **F** Battenberg near the *CANT1* locus on chromosome 17 in two samples from patient A10 (two other samples from this patient are shown in Additional file 1: Fig. S4)

to the published HATCHet results [30]. The full haplotype-specific copy-number profiles inferred by HATCHet2 for all patients are shown in Additional file 1: Fig. S13.

We first observe that the copy-number segments inferred by the three methods have very different length distributions. Battenberg produces a small number of large segments; HATCHet produces a large number of small segments; and HATCHet2 produces a balance of large and small segments (Fig. 2A). This difference reflects the methodological approaches of the three methods: Battenberg identifies copy-number segments separately in each sample, strongly relying on local segmentation across the genome; HATCHet employs a global clustering approach to group genomic bins across the genome and across samples, ignoring the location of bins on the genome. In contrast, HATCHet2 employs a *local-global* clustering algorithm that uses both

local information along the genome as well as global clustering of bins across samples (see the "Methods" section for details). Thus, HATCHet2 identifies more clone-specific focal CNAs than Battenberg, while avoiding the over-fragmentation of copy-number segments observed in HATCHet.

We next assessed whether the differences in copy-number segments among the methods affected known cancer genes. We focused on regions near 41 genes highlighted in The Cancer Genome Atlas (TCGA) prostate cancer publication [72] (we include in Additional file 1: Fig. S2 a similar analysis of COSMIC [73] Cancer Gene Census (CGC) genes). We counted the number of copy-number segments reported within 1 megabase of each gene (Fig. 2B). Consistent with the results for the length distribution of segments, we observed that Battenberg yields only a single segment spanning the entire region for 367/410 gene-patient pairs (89.5%). In contrast, HATCHet often has many segments surrounding these genes: up to 21 segments within 1 Mb from *ATM* in patient A29. HATCHet2 infers fewer segments than HATCHet, producing simpler copy-number profiles, but infers more segments than Battenberg, which can better fit the data when there is evidence of copy-number change. To demonstrate the advantages of HATCHet2 and their impact, we focus in detail on two examples involving genes with previously reported roles in prostate cancer: *TP53* [74, 75] and *CANT1* [76, 77].

We further examined the genomic region containing *TP53* in 3 bulk tumor samples from patient A12 where there were considerable differences between the HATCHet2 and HATCHet copy-number profiles. Specifically, in the 10-Mb region centered at *TP53*, HATCHet2 identifies 3 copy-number segments among 3 tumor clones in the region, with the segment containing *TP53* having clonal LOH (Fig. 2C and Additional file 1: Fig. S3B). In contrast, HATCHet reported 35 segments in this region (Fig. 2D and Additional file 1: Fig. S3A), with segments switching between two distinct copy-number states: clonal LOH (orange) and subclonal LOH (red). Due to these switches in copy-number states across different clones, HATCHet reports that the segment containing *TP53* is a region with a subclonal LOH, compared to HATCHet2's report of clonal LOH. The HATCHet2 result is consistent with recent pan-cancer studies which report that *TP53* is commonly lost or inactivated early during cancer progression and appears as a clonal copy-number event [78]. Further supporting HATCHet2's inference of clonal LOH at the *TP53* locus, the original publication [42] reported that this patient contains the *TP53* missense variant rs1800371 G>A which has been shown to impair p53 function when only one mutated allele is retained [79].

As another example, we examined the genomic region overlapping the *CANT1* gene in 4 bulk tumor samples from patient A10, where we observed substantial differences between the HATCHet2 and Battenberg copy-number profiles. Specifically, in the 5-Mb region around the *CANT1* gene, HATCHet2 identifies 6 segments across the tumor samples that closely match the observed sequencing data (Fig. 2E and Additional file 1: Fig. S4B). In contrast, Battenberg subdivides the region into 2–5 segments across the 4 samples but fails to identify several obvious focal copy-number aberrations (Fig. 2F and Additional file 1: Fig. S4C). In fact, in all but sample A10-A, Battenberg does not accurately fit the copy-number state at the *CANT1* locus in terms of mean squared error (MSE): the HATCHet2 solution has a BAF MSE of 0.0068 and

RDR MSE of 0.78 across the 3 remaining samples, whereas the Battenberg solution has a BAF MSE of 0.0090 and RDR MSE of 1.53.

Overall, on this dataset, we find that HATCHet2 identifies focal CNAs that are missed by Battenberg-due to Battenberg's strong reliance on segmentation of individual samples-while avoiding the many spurious focal CNAs reported by HATCHet-due to HATCHet's global clustering that ignores correlations between copy-number states on the genome. We further characterize the focal CNAs inferred by the three methods in Additional file 1: Section S10.

### Identification of clone-specific mirrored-subclonal CNAs

A key feature of HATCHet2 is the identification of clone-specific mirrored-subclonal CNAs in multi-region or multi-sample sequencing data. Mirrored-subclonal CNAs are differential gains or losses of the maternal and paternal chromosomes in different subpopulations of cancer cells from the same tumor. In contrast to earlier reports of mirrored-subclonal allelic imbalance [39, 48–50] in bulk samples, HATCHet2 identifies haplotype imbalance in specific clones. We show using simulated data that HATCHet2 accurately identifies mirrored-subclonal CNAs (Additional file 1: Section S9). Our HATCHet2 analysis of the 10 prostate cancer patients from Gundem et al. [42] reveals numerous mirrored-subclonal CNAs that were missed in both the original published analysis of these patients [42] as well as in the reanalysis using HATCHet [30]. In particular, across these patients, HATCHet2 identified 57 mirrored-subclonal CNAs overlapping TCGA prostate cancer genes (Fig. 3A).

To further contrast the results from HATCHet2 with the previously published results from HATCHet and Battenberg on this dataset, we examined the mirrored-subclonal CNAs inferred by each method. Across the 10 patients, HATCHet2 infers an average of 204 segments per patient where at least one clone has a mirrored-subclonal CNA, present in 18.2% of the tumor genome on average, and overlapping an average of 5.5 TCGA prostate cancer genes in each patient. HATCHet can infer mirrored-subclonal CNAs since it uses the same factorization as HATCHet2, but because the mBAF does not respect the invariance of haplotype across samples, it is much less informative than the mhBAF used in HATCHet2. As a result, HATCHet identifies mirrored events in only 5.5% of the segments in which HATCHet2 identifies a mirrored event. Battenberg is incapable of inferring mirrored-subclonal CNAs, as it will never return a copy-number state $(A, B)$ where $B > A$.

We focus on patient A10 with 4 tumor samples. In this patient, HATCHet2 identified 4 tumor clones and revealed mirrored-subclonal CNAs in 162/2180 segments (7.4%), representing 4.2% of the genome. In particular, HATCHet2 found that clone 2, which makes up the entirety of tumor cells in sample A10-A, had copy-number state (1,2) or (2,3) across regions of chromosomes 1 and 2 totaling 40.5 Mb (Fig. 3B–C, dotted blue boxes). Within these same regions, clones 3 and 4 had amplifications of the *opposite* allele with copy-number states (2,1), (3,1), (4,1), and (4,2) (not shown).

The mirrored-subclonal CNAs identified by HATCHet2 overlap 6 prostate cancer genes in COSMIC [73]: *ELK4*, *SLC45A3*, *HNRNPA2B1*, *SPOP*, *KLK2*, and *CANT1*. In particular, genes *ELK4* and *SLC45A3* are known to form a chimeric transcript that is involved in cell proliferation in prostate cancer [52, 53]. To provide further support for

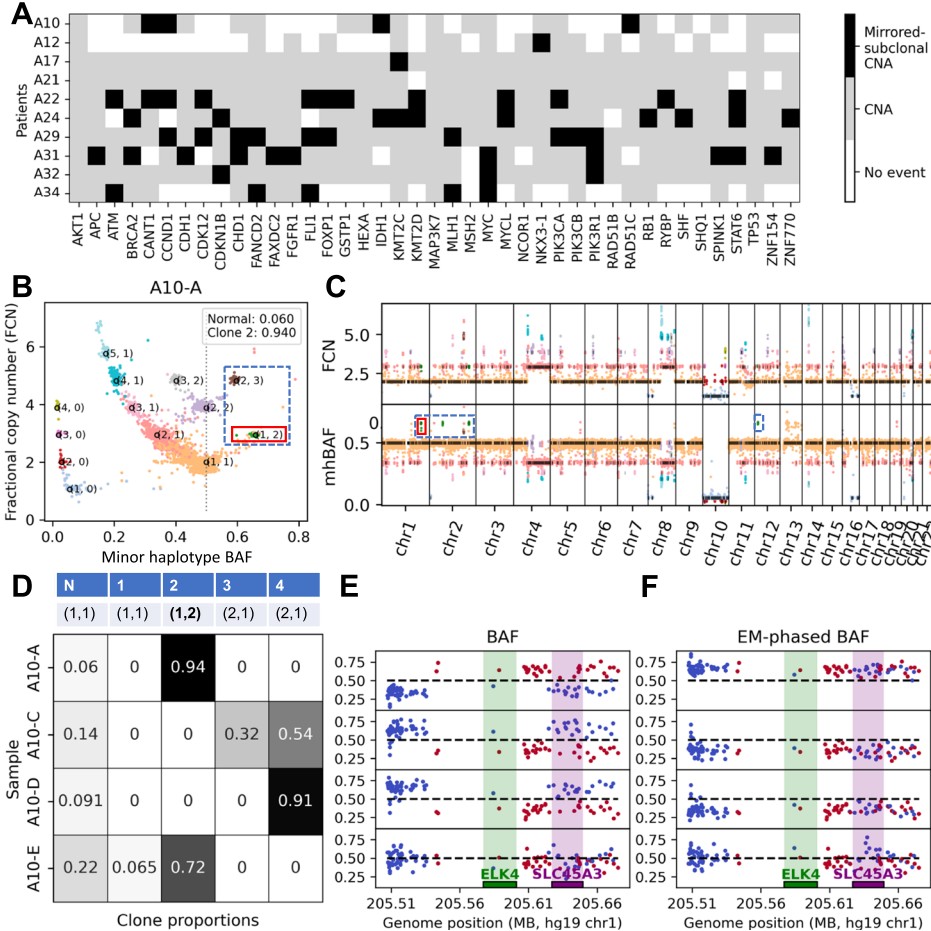

**Fig. 3** Mirrored-subclonal copy-number aberrations in prostate cancer identified by HATCHet2. **A** Copy-number landscape for 41 prostate cancer genes from The Cancer Genome Atlas [72] across 10 prostate cancer patients from Gundem et al. [42]. White entries indicate that no copy-number aberration was identified at the locus. Black entries indicate that mirrored-subclonal CNAs were observed among the tumor clones at the locus. Gray entries indicate that a non-mirrored copy-number aberration was inferred for at least one clone at the locus. **B** Inferred copy-number states for the single tumor clone present in prostate cancer sample A10-A. Each point is a genomic bin whose position corresponds to its inferred minor haplotype BAF (mhBAF, *x*-axis) and fractional copy number (FCN, a rescaling of the read-depth ratio, *y*-axis). Each point is colored by the copy-number state assigned to the bin. Points labeled (*a, b*) are the expected position of the corresponding haplotype-specific copy-number state with *a* copies of the major haplotype and *b* copies of the minor haplotype. Dotted blue box indicates mirrored-subclonal CNAs, and red box indicates the mirrored-subclonal CNA examined in panels **D**–**F**. **C** Fractional copy number (top) and mhBAF (bottom) values across the genome. Black lines indicate the expected FCN and mhBAF of the assigned copy-number state (analogous to labeled points in panel **B**). Dotted blue boxes indicates mirrored-subclonal CNAs, and red box indicates the mirrored-subclonal CNA examined in panels **D**–**F**. Points are colored by the assigned haplotype-specific copy-number state as in **B**. **D** Inferred haplotype-specific copy numbers (*a, b*) (first row) and clone proportions (entries in table) for the normal clone (N) and 4 tumor clones (1–4) for the segment containing the genes *ELK4* and *SLC45A3*. **E** BAF values (i.e., the fraction of reads with the non-reference allele) across samples for SNPs in the bin containing genes *ELK4* (green bar) and *SLC45A1* (purple bar). Blue points indicate SNPs that have BAF $\leq$ 0.5 in sample A10-A, while red points indicate SNPs with BAF > 0.5 in sample A10-A. Note that in samples A10-C and A10-D, the blue and red points are reflected across the dotted line at BAF = 0.5 relative to sample A10-A, indicating mirrored-subclonal CNA. **F** Haplotype-phased BAF values (i.e., either the fraction of alternate reads or the fraction of reference reads as indicated by the phasing inferred by HATCHet2) across samples for SNPs in the bin containing genes *ELK1* and *SLC45A1*. SNPs are colored as in panel **E**. Note that SNPs of different colors (i.e., different BAF values in A10-A) have been grouped together via HATCHet2's mhBAF inference algorithm to show that the haplotype containing these SNPs is more abundant in sample A10-A (phased BAFs > 0.5) but less abundant in samples A10-C and A10-D (phased BAFs < 0.5)

the mirrored-subclonal amplification containing these cancer-indicated genes, we examined the segment containing *ELK4* and *SLC45A3*. Figure 3D shows the haplotype-specific copy numbers (top) for each of the 4 tumor clones in this segment, as well as their clone proportions (bottom) in the 4 samples (rows). While the observed BAFs for the heterozygous germline SNPs in this segment ranged from 0.14 to 1 (Fig. 3E), the mhBAF inference algorithm in HATCHet2 grouped these SNPs together to reveal that the more abundant haplotype in sample A10-A was different from the more abundant haplotype in the remaining samples (Fig. 3F). While gene fusions are canonically driven by genomic translocations [80] and we do not see evidence of a deletion between *SLC45A3* and *ELK4* (i.e., mhBAF closer to 0.5 than in the neighboring regions), the fusion transcript of these adjacent genes has been previously reported to result from *cis*-splicing [53]. Thus, the amplification of these genes could be related to expression of this fusion transcript despite the lack of direct genomic evidence. Additionally, both the primary tumor sample A10-E and the right iliac lymph node metastasis A10-A consist mainly of clone 2 with amplifications of the minor haplotype, while the perigastric lymph node metastasis A10-C and periportal lymph node metastasis A10-D consist of clones 3-4 which have amplifications of the major haplotype in this region (Fig. 3D). This suggests that there may have been two distinct migration events from the primary tumor and that the right iliac lymph node that shares a copy-number clone with the primary tumor may have been seeded much later than the perigastric and periportal lymph nodes whose clones are distinct from the tumor clone in the primary tumor sample. Overall, the identification of these mirrored-subclonal amplifications provides evidence of parallel evolution for the amplification of these genes.

### Validating HATCHet2 using single-cell whole-genome sequencing data

We further validated HATCHet2's ability to identify tumor subclones by analyzing pseudobulk data created from single-cell whole-genome DNA sequencing data. Recent single-cell whole-genome DNA sequencing technologies [19, 54, 81–85] provide measurements of CNAs in individual cells, but often with low coverage per cell, limiting the resolution of CNA identification. We analyzed a single-cell whole-genome sequencing dataset from the 10x Genomics CNV solution [54], which contains ≈0.03X coverage from ≈10,000 single cells obtained from five spatial sections of a breast tumor. Allele- and haplotype-specific CNAs were previously inferred on this dataset using the CHISEL algorithm [15]. This dataset serves as a useful benchmark since the single-cell analysis identified the presence of two major tumor clones with two large mirrored-subclonal CNAs affecting the entirety of chromosomes 2 and 3. Thus, the dataset is a useful test of HATCHet2's ability to identify clone-specific mirrored-subclonal CNAs.

We ran HATCHet2 on four pseudobulk samples from sections B–E of the tumor; each pseudobulk sample was generated by aggregating all reads from all cells of the corresponding section. In total, these four pseudobulk samples consisted of: 40.9X coverage from 2224 cells (section B), 37.3X coverage from 1722 cells (section C), 33.6X coverage from 1915 cells (section D), and 44.1X coverage from 2053 cells (section E). As in previous analysis [15], we treated the low-tumor-purity section A as a matched normal sample and applied HATCHet2 to identify tumor clones in the remaining four sections B–E.

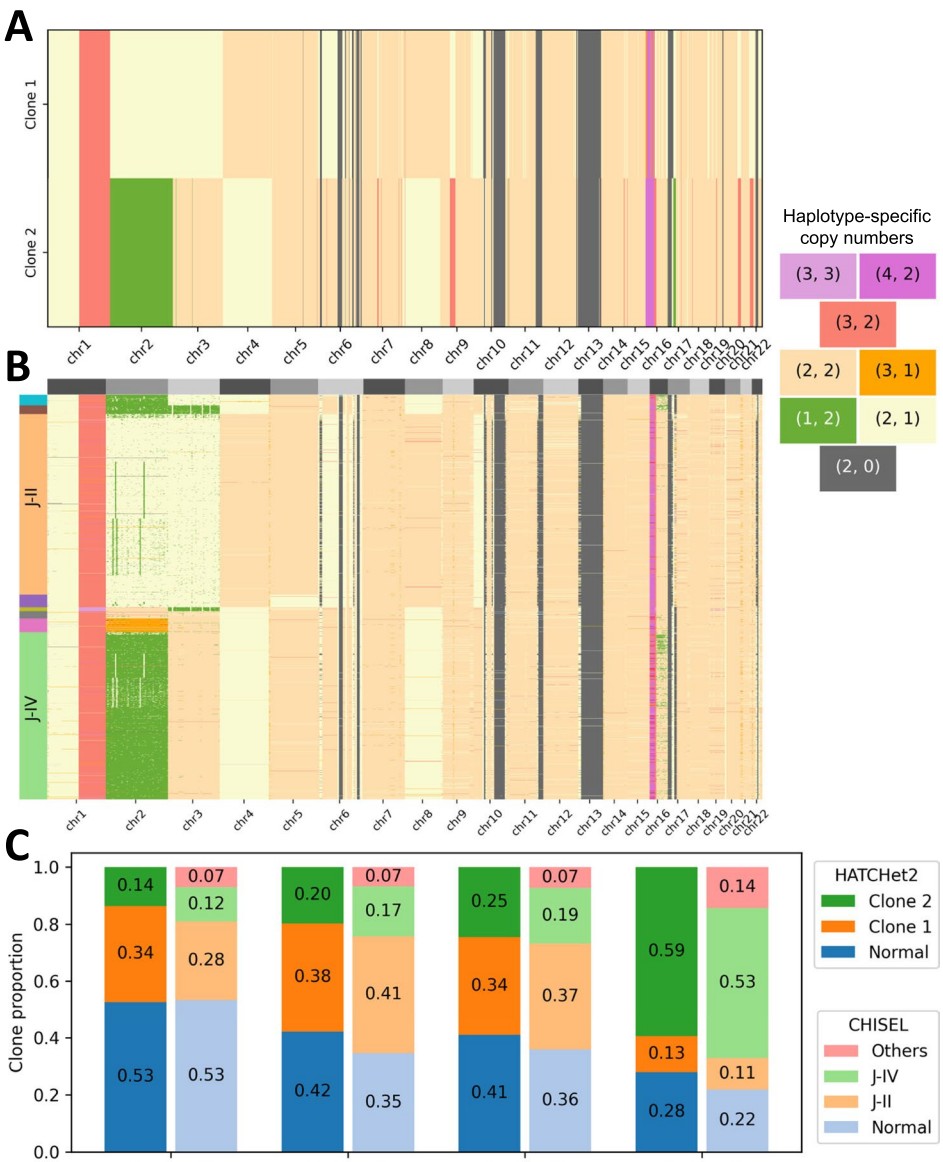

**Fig. 4 A** Haplotype-specific copy-number profiles for two tumor clones identified by HATCHet2 on 4 pseudobulk samples from single-cell whole-genome sequencing data from 7914 cells of 4 sections a breast tumor. Each copy-number segment is colored by the assigned haplotype-specific copy-number state (labeled in the legend on the right). **B** Single-cell haplotype-specific copy-number profiles identified by CHISEL [15]. Rows correspond to cells and are grouped into copy-number clones, which are annotated on the left. Each entry is colored by its assigned haplotype-specific copy-number state, as in panel **A**. **C** Clone proportions inferred by HATCHet2 (left bar in each pair) and CHISEL (right bar in each pair) in each sample (*x*-axis)

HATCHet2 identified 2 tumor clones across the pseudobulk samples (Fig. 4A). The copy-number profiles of these two clones closely match the copy-number profiles of the two major clones inferred by CHISEL (comprising ≈ 86% of the tumor cells) on the single-cell sequencing data (Fig. 4B). In particular, we found that the HATCHet2 clone 1 haplotype-specific copy-number profile matches the CHISEL single-cell clone J-II copy-number profile across 96% of the genome. Similarly, HATCHet2 clone 2 shares 94%

of its haplotype-specific copy-number profile with the CHISEL single-cell clone J-IV (Additional file 1: Fig. S6). The two copy-number clones identified by HATCHet2 capture the large subclonal copy-number aberrations that differ between CHISEL clones J-II and J-IV, including the mirrored-subclonal amplification on chromosome 2. HATCHet2 agrees with the CHISEL single-cell analysis regarding the subclonal differences on chromosomes 3, 4, 6, 8, 9q, and 10 as well as large clonal events including chromosome 1p deletion, chromosome 1q amplification, LOH regions on chromosome 10, chromosome 11, and chromosome 13, and multi-copy amplifications on chromosome 16. Of course, given the limitations of bulk sequencing compared to single-cell, there are some CNAs that are missed by HATCHet2. For example, HATCHet2 does not recover the mirrored-subclonal event on chr3 that affects only 3.3% of cells. These differences are not surprising since these CNAs are extremely weak signals in the pseudobulk data.

HATCHet2 also identifies the presence of a whole-genome duplication in both tumor clones, in agreement with the clonal whole-genome duplication identified by CHISEL in the published single-cell analysis [15]. Finally, the proportions of these two clones in the four sections is highly concordant between the HATCHet2 pseudobulk analysis and the CHISEL single cell analysis (mean absolute difference 0.040; total variation distance in each sample $\leq 0.0715$), providing further evidence to the accuracy of HATCHet2's results (Fig. 4C).

## Discussion

We introduced HATCHet2, an algorithm to infer clone-specific CNAs from DNA sequencing data from one or more bulk tumor samples from the same patient. HATCHet2 includes several improvements over previous methods, including a novel quantity called the minor haplotype BAF (mhBAF) that enables the inference of clone-specific mirrored-subclonal CNAs and local-global clustering of dynamic-width genomic bins that enables the inference of clone-specific focal CNAs. The combination of these improvements allows HATCHet2 to outperform several previous methods on simulated data. On a previously published dataset of multi-region sequencing from 10 metastatic prostate cancers, we demonstrated that HATCHet2 retrieves more realistic focal CNAs and reveals previously unknown mirrored-subclonal CNAs. Both types of events are well supported by the underlying DNA sequencing data and affect important genes in prostate cancer. Importantly, we also validated the accuracy of HATCHet2 in identifying mirrored-subclonal CNAs using simulated data and single-cell DNA sequencing datasets.

There are a few areas where HATCHet2 could be further improved. First, the calculation of RDR for a bin may be improved by taking into account specific characteristics of each genomic region such as GC bias [86] and mappability [87]. Second, the greedy algorithm for variable-width binning might be improved by formulating an exact optimization problem. Third, the local-global clustering method can be improved by using a probabilistic model for read count data instead of modeling the RDR and mhBAF with normal distributions. Finally, while HATCHet2 provides high-accuracy results in practice, there are a few user-tuneable parameters including the minimum number of SNP-covering and total reads per bin and the maximum integer copy-number state that might

benefit from automated model selection procedures, particularly on challenging datasets with low coverage and/or low tumor purity.

## Conclusions

DNA sequencing of bulk tumor samples remains the most common technique used in cancer research studies to investigate somatic intra-tumor heterogeneity and cancer evolutionary histories [14, 88]. Thus, better understanding of cancer evolution relies on the accuracy of deconvolution methods that identify genomic alterations-including CNAs-in bulk sequencing data and assign these alterations to distinct tumor clones. We demonstrated that the HATCHet2 algorithm introduced here provides key improvements to CNA identification from bulk data and to CNA assignment to tumor clones. HATCHet2 improves the inference of clone-specific CNAs, as demonstrated in this study using simulations and single-cell sequencing datasets. Furthermore, HATCHet2 identifies important cancer events, such as clone-specific focal CNAs and clone-specific mirrored-subclonal CNAs, that are often missed by previous methods. We thus expect that the application of HATCHet2 to large multi-region datasets from different cancer types will provide novel insights into cancer evolutionary processes, including parallel evolution via mirrored-subclonal CNAs and the clonal relationships between different regions of a primary tumor or between a primary tumor and multiple metastases.

## Methods

### Overview of HATCHet2

HATCHet2 infers haplotype-specific copy-number profiles and cellular proportions of multiple distinct tumor clones from DNA sequencing data of one or more bulk tumor samples from a cancer patient. Specifically, the inputs to HATCHet2 are aligned DNA sequencing reads (in the form of BAM files [89]) obtained from one or more tumor samples of an individual patient as well as the aligned DNA sequencing reads obtained from a matched normal sample (e.g., from blood or adjacent normal tissue).

HATCHet2 assumes that a tumor is composed of $m$ distinct tumor clones, where each tumor clone corresponds to a population of cancer cells with the same copy-number aberrations (CNAs). Each of these CNAs alters the number of copies of one of the two homologous chromosomes, labeled as $\mathcal{A}$ and $\mathcal{B}$, of the normal diploid human genome. These CNAs divide the genome into $n$ genomic bins, where each bin corresponds to a contiguous segment of the genome that is affected by the same CNAs. Thus, each tumor clone $i$, $1 \leq i \leq m$, corresponds to a pair $(\mathbf{a}_i, \mathbf{b}_i)$ of integer vectors of length $n$ whose entries $a_{i,s}$ and $b_{i,s}$ indicate the number of copies of the $\mathcal{A}$ and $\mathcal{B}$ allele, respectively, that are present in the genome of clone $i$ at genomic bin $s$. We define the normal clone $i = 0$ as the clone composed of diploid cells with copy number $a_{0,s} = b_{0,s} = 1$ for both alleles of every genomic bin $s$.

Each bulk sample is a mixture of normal cells and cells from one or more tumor clones. We define the *clone proportion* $u_{i,p}$ to be the proportion of cells in sample $p$ that belong to clone $i$. Thus, we have that $\sum_{i=0}^{m} u_{i,p} = 1$, and $1 - u_{0,p}$ corresponds the the *tumor purity* of sample $p$, or the proportion on non-normal cells in sample $p$.

HATCHet2 uses two summary statistics obtained from the DNA sequencing reads aligned to each genomic bin $s$ to identify the underlying CNAs: the read-depth ratio

(RDR) and the B-allele frequency (BAF). The RDR $r_{s,p}$ is the ratio between the number of aligned reads whose starting position is in bin $s$ in tumor sample $p$ and the corresponding number of reads in the matched normal sample. The RDR $r_{s,p}$ is proportional to the *total* number of copies of bin $s$ in sample $p$. The BAF $\beta_{j,p}$ of a single-nucleotide polymorphism (SNP) $j$ in sample $p$ is the fraction of reads overlapping SNP $j$ that contain the non-reference allele. The BAF at heterozygous germline SNP positions is proportional to the *allelic imbalance*, i.e., the relative abundance of $\mathcal{A}$ and $\mathcal{B}$ copies, at each SNP.

The goal of HATCHet2 is to infer haplotype-specific copy-number profiles $(\mathbf{a}_i, \mathbf{b}_i)$ for $i = 1, \ldots, n$ clones and clone proportions $u_{i,p}$ for each cancer clone $i$ in each tumor sample $p$, from DNA sequencing reads from multiple bulk tumor samples. HATCHet2 consists of five steps.

1. *Genotyping and reference-based phasing of germline SNPs.* HATCHet2 extracts the read coverage at genomic locations from the dbSNP database of common human SNPs [55] and identifies heterozygous germline SNPs using the matched normal sample. Depending on sequencing coverage, individual SNPs may give poor estimates of the BAF. Thus, following the procedure in CHISEL [15], we optionally phase adjacent SNPs into haplotype blocks using a reference-based phasing algorithm [56] and a panel of haplotypes from thousands of human genomes [90]. We then aggregate counts for the phased SNPs in each local haplotype block into a single BAF estimate.

2. *Variable-width binning.* HATCHet2 divides the genome into *variable-width bins* in which each bin has at least $k_{SNP}$ SNP-covering reads and $k_{total}$ total reads in each sample. The variable-width binning procedure helps stabilize the variance across bins. Both $k_{SNP}$ and $k_{total}$ are user-customizable parameters.

3. *Calculation of read-depth ratio (RDR) and minor haplotype B-allele frequencies (mhBAF).* For each bin, we compute the read-depth ratio (RDR) and a quantity called the *minor haplotype B-allele frequency (mhBAF)* for each sample. The mhBAF of a bin is the frequency of the haplotype with the lower average abundance across samples. Computing the mhBAF for a bin requires phasing the SNPs in a bin into two parental haplotypes-we perform this phasing and estimate the mhBAF using an EM algorithm.

4. *Cluster bins into copy-number states across the genome and across samples using local-global clustering.* While most previous approaches adopt either an exclusively local (e.g., piecewise segmentation [35, 36]) or exclusively global (e.g., GMM clustering [30]) algorithm to group bins into copy-number segments, we introduce a hybrid approach that clusters bins across the genome and across samples while leveraging local information.

5. *Compute haplotype-specific copy numbers and clone proportions.* HATCHet2 solves a mixed integer factorization problem to deconvolve the RDR and mhBAF values of each cluster into integer haplotype-specific copy-number profiles $(\mathbf{a}_i, \mathbf{b}_i)$ for each clone $i$ and clone proportions $u_{i,p}$ for each clone $i$ in each sample $p$. HATCHet2 selects the number of clones and whole-genome duplication status using a model selection procedure.

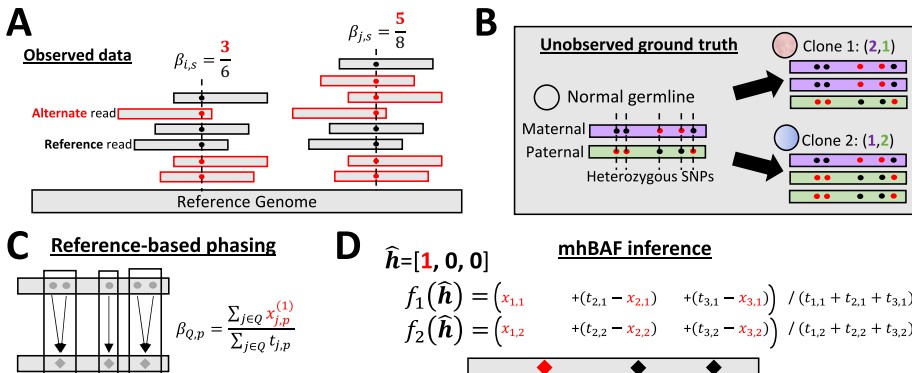

**Fig. 5** Inference of minor haplotype BAF (mhBAF). **A** The BAF $\beta_{j,p}$ for SNP $j$ and sample $p$ is the fraction of reads covering the SNP that contain the alternate allele (red) and is a signal for allelic imbalance. **B** Positions of five heterozygous SNPs, with alternate alleles given in red, for a bin with a mirrored-subclonal copy-number aberration, with distinct haplotype-specific copy-number states in two tumor clones. **C** Reference-based phasing groups together SNPs that are likely to be on the same haplotype based on a reference database. We combine read counts for each of these haplotype blocks to obtain a single BAF estimate. **D** The minor haplotype BAF (mhBAF) is estimated by inferring a phasing $\hat{h}$ that assigns alternate alleles to a haplotype, and computing the frequency $f_p(\hat{h})$ of this haplotype in each sample $p$

Step 5 is methodologically unchanged between HATCHet and HATCHet2. In the following sections, we describe how HATCHet2 performs steps 1–4.

### Genotyping and reference-based SNP phasing

Heterozygous germline SNPs are positions at which the two parental alleles differ, and these SNPs distinguish between the two haplotypes in the tumor genome (Fig. 5A–B). We identify these heterozygous germline SNPs from the matched normal sample using a reference list of common SNPs (dbSNP [55] by default) and `bcftools call` [91].

Depending on the sequencing coverage, the BAF at an individual SNP may provide a poor estimate of the allelic imbalance. Ideally, one would aggregate counts from multiple nearby SNPs, but one cannot do this without knowing the phase of each SNP, i.e., which allele is maternal vs. paternal. Due to human population structure and limited recombination, some haplotypes occur more commonly than others. Reference-based phasing methods [56, 56, 57, 92] exploit this structure using a large database of known haplotypes to identify SNPs that are likely to be grouped together. Following the approach in several previous CNA inference methods [15, 22, 23, 26, 66], HATCHet2 applies a reference-based phasing algorithm to infer the phase of alleles across whole chromosomes. Since the accuracy of phasing decreases with genetic distance [93], we use the inferred phase to group together SNPs into haplotype blocks with a user-defined maximum length (25 kilobases by default).

Reference-based phasing gives only an approximation of the haplotype phase, and *switch errors*-erroneously switching phase between parental haplotypes-occur with appreciable rates [93]. These errors may be due to errors in the reconstructed haplotypes within the reference panel (particularly for rare alleles [94]) or because the target individual may have a novel haplotype not present in the panel. Indeed, reference-based phasing may be less accurate for individuals from populations that are poorly represented in the reference database [95]. To avoid propagating switch errors, HATCHet2

includes the following statistical test. Let $x_i$ indicate the number of reads at SNP $i$ that are indicated by the reference phasing to contain the allele on the "1" haplotype, and let $y_i$ indicate the corresponding value for the "0" haplotype. For each pair $i, j$ of adjacent SNPs, if the phased allele frequency $\frac{x_i}{(x_i+y_i)}$ is significantly different from $\frac{x_j}{(x_j+y_j)}$, which may occur when there are switch errors in regions with allelic imbalance, then we do not combine read counts between SNPs $i$ and $j$. Instead, we defer the grouping of SNPs to the subsequent steps of HATCHet2.

This approach is implemented in an automatic phasing pipeline in HATCHet2. Further details on the phasing approach are in Additional file 1: Section S3. Note that Numbat [96], a recent method for CNA inference from single-cell data, also aims to correct errors in reference-based phasing, albeit as part of a much more complex model.

In subsequent steps of HATCHet2, we represent each set $Q$ of heterozygous SNPs grouped together by the reference-based phasing pipeline as a single "meta-SNP" with non-reference counts $x_{Q,p} = \sum_{j \in Q} x_{j,p}^{(1)}$ and total counts $t_{Q,p} = \sum_{j \in Q} t_{j,p}$, where $x_{j,p}^{(1)}$ indicates the number of reads at SNP $j$ in sample $p$ assigned to haplotype "1" by reference-based phasing. Thus, the BAF $\beta_{Q,p} = \frac{\sum_{j \in Q} x_{j,p}^{(1)}}{\sum_{j \in Q} t_{j,p}}$ (Fig. 5C).

**Variable-width binning**

The next step of HATCHet2 is to divide the genome into distinct non-overlapping sections called *bins*, which are the smallest regions where integer copy number will be computed. For each bin, HATCHet2 computes two statistics, the RDR and the mhBAF, both of which depend on the number and type of reads that align to the bin. Specifically, let $d_{s,p}$ be the total number of reads aligned to bin $s$ in sample $p$ (where $p = 0$ represents the matched normal sample). Similarly, for each heterozygous germline SNP $j$, let $x_{j,p}$ be the number of alternate (non-reference) reads in sample $p$ and let $t_{j,p}$ the total reads overlapping SNP $j$ in sample $p$. The RDR $r_{s,p} = \frac{d_{s,p}}{d_{s,0}}$ depends on the total number of reads that align to each bin, which is proportional to the width of the bin and the number of copies of the corresponding genomic region in each clone in the sample. However, the BAF $\beta_{j,p} = \frac{x_{j,p}}{t_{j,p}}$ depends on the number $t_{j,p}$ of reads that overlap heterozygous germline SNP positions.

Early copy-number inference methods-many designed for SNP array or targeted sequencing data-compute RDR and BAF at individual SNP positions before identifying copy-number segments [22–24, 28, 31, 34, 35]. Later methods for DNA sequencing data typically divide the genome into *fixed-width bins* and call copy number at each bin by counting sequencing reads that fall within each bin [20, 25, 26, 29, 87, 97, 98]. Since heterozygous SNPs are not uniformly distributed across the genome [99, 100], methods that aggregate alternate read counts from multiple heterozygous SNPs in fixed-width bins-including HATCHet [30]-may have large variation in the total number $\sum_j t_{j,p}$ of SNP-covering reads and thus high variance in the BAF across bins with the same allelic imbalance.

To illustrate this phenomenon, consider the BAF of a single meta-SNP $Q$. The BAF $\beta_{Q,p} = \frac{\sum_{j \in Q} x_{j,p}^{(1)}}{\sum_{j \in Q} t_{j,p}}$ of meta-SNP $Q$ is the maximum likelihood estimator of the fraction of $\mathcal{B}$ alleles in the sample. The variance of this estimator depends on the total number $t_Q = \sum_{j \in Q} t_{j,p}$ of SNP covering reads in the meta-SNP. With fixed-width bins, $t_Q$ will vary

considerably across the genome following the variability in SNP density across the genome. To address this issue, HATCHet2 divides the genome into *variable-width* bins to stabilize the variance of the allelic imbalance signal. Similar variable-width binning strategies for total read counts rather than SNP-covering reads have been used for identifying copy-number aberrations in single-cell DNA sequencing data [17, 18, 101, 102] to stabilize the expected read depth due to varying mappability across the genome.

HATCHet2 includes a greedy *variable-width* binning algorithm that ensures that each bin has at least $k_{SNP}$ SNP-covering reads and at least $k_{total}$ total reads in each sample. This algorithm walks along a chromosome arm counting the number of SNP-covering and total reads, and places a bin boundary once these accumulated counts exceed $k_{SNP}$ and $k_{total}$, respectively, in all samples. Once the end of the chromosome arm is reached, the last bin boundary that was placed is removed to ensure that all bins have sufficiently many counts. If there are not enough counts across an entire chromosome arm, the arm is treated as a single bin. Overall, this algorithm aims to maximize the number of bins (i.e., minimize the average bin width) subject to the constraints on minimum total and SNP-covering reads.

After binning, HATCHet2 computes the read-depth ratio and normalizes it to account for differences in coverage between samples. Specifically, for each bin $s$ and tumor sample $p$, we compute the raw read-depth ratio $\tilde{r}_{s,p}$ by dividing the total number $d_{s,p}$ of reads starting in bin $s$ in sample $p$ by the number $d_{s,0}$ of reads starting in bin $s$ in the matched normal sample $p_0$. We then apply library size normalization to control for the total number of reads in tumor sample $p$ and the matched normal sample: $r_{s,p} = \tilde{r}_{s,p} \cdot \left( \frac{\sum_s \tilde{r}_{s,p_0}}{\sum_s \tilde{r}_{s,p}} \right)$. Thus, $r_{s,p}$ values are comparable between tumor samples $p$ up to differences in the tumor purity and clone proportions in those samples.

### Minor haplotype B-allele frequency (mhBAF)

To estimate the haplotype-specific copy number for each bin, we first estimate a single measure of "allelic imbalance" for the bin, i.e., the relative abundance of the "B-allele" of the bin. For a bin containing multiple germline SNPs, quantifying the "B-allele" requires phasing the alleles of all heterozygous germline SNPs in the bin into the two parental haplotypes. For bins of larger width (i.e., tens to hundreds of kb), reference-based phasing can be unreliable [93]. We define the *minor haplotype B-allele frequency (mhBAF)* $\hat{f}_{p,s}$ of bin $s$ in sample $p$ to be the frequency in sample $s$ of the haplotype for bin $s$ with lower average abundance across samples.

We compute the mhBAF for a bin $s$ as follows. Let $\mathbf{h} = (h_1, h_2, \ldots h_\ell)$ be a vector of the (unobserved) phases of the $\ell$ SNPs in a bin, where $h_j = 1$ if the alternate allele for SNP $j$ is on one particular parental haplotype, and $h_j = 0$ if the alternate allele for SNP $j$ is on the *other* haplotype. Let $x_{j,p}$ be the number of alternate (non-reference) reads for SNP $j$ in sample $p$ and $t_{j,p}$ be the corresponding total number of reads (Fig. 5A).

Given the haplotype phasing $\boldsymbol{h}$, one can estimate the frequency $f_p(\boldsymbol{h})$ of this haplotype in sample $p$ directly:

$$f_p(\boldsymbol{h}) = \frac{\sum_{j=1}^{\ell} h_j x_{j,p} + (1 - h_j)(t_{j,p} - x_{j,p})}{\sum_{j=1}^{\ell} t_{j,p}}. \tag{1}$$

$f_p(\boldsymbol{h})$ is the maximum-likelihood estimate of the proportion of copies of haplotype $\boldsymbol{h}$ in sample $p$.

However, we do not observe $\boldsymbol{h}$ – rather, we only observe the read counts $x_{j,p}$ and $t_{j,p}$. Using these counts, we identify those SNPs that appear in similar frequency across samples. Let $\hat{\beta}_{j,p}(h_j) = \frac{h_j x_{j,p} + (1-h_j)(t_{j,p} - x_{j,p})}{t_{j,p}}$ represent the phased BAF of SNP $j$ in sample $p$ with phase $h_j$. In particular, we expect to observe exactly two sets of SNPs: one set whose alternate alleles are on one parental haplotype $\boldsymbol{h}$ with phased BAFs $\hat{\beta}_{j,p}(h_j) \approx f_p(\mathbf{h})$ for all $j$, $p$; and the other set whose alternate alleles are on the other parental haplotype $\overline{\boldsymbol{h}} = (1 - h_1, 1 - h_2, \ldots, 1 - h_\ell)$ with phased BAFs $\hat{\beta}_{j,p}(\overline{h}_j) \approx f_p(\overline{\boldsymbol{h}}) = 1 - f_p(\boldsymbol{h})$ for all $j$, $p$ (Fig. 5D). Because we cannot in practice differentiate whether a haplotype is maternal or paternal, the order between these two sets of SNPs does not matter: $\boldsymbol{h}$ and its complement $\overline{\boldsymbol{h}}$ yield the same unordered set $\{f_p(\boldsymbol{h}), 1 - f_p(\boldsymbol{h})\}$ of haplotype frequencies. To resolve this ambiguity, we select the vector $\hat{\boldsymbol{h}}$ that produces smaller haplotype frequencies on average across samples:

$$\hat{\boldsymbol{h}} = \underset{\boldsymbol{h}' \in \{\boldsymbol{h}, \overline{\boldsymbol{h}}\}}{\operatorname{argmin}} \frac{1}{P} \sum_p f_p(\boldsymbol{h}') \tag{2}$$

We refer to $\hat{f}_p = f_p(\hat{\boldsymbol{h}})$ as the *minor haplotype B-allele frequency* (mhBAF), as it is an estimate of the frequency of the parental haplotype that is less abundant on average across samples. Moreover, this definition of the mhBAF implies that $\hat{f}_p \leq 0.5$ in most samples $p$, except in samples where the less abundant haplotype *across* all samples is more abundant in a particular sample; such cases correspond to mirrored-subclonal allelic imbalance [39] and indicate mirrored-subclonal CNAs. We simultaneously estimate the SNP phasing $\hat{\boldsymbol{h}}$ and the mhBAF $\hat{f}_p$ by adapting an EM algorithm originally applied to single-cell haplotype frequency estimation [66], which extends an earlier EM algorithm used to phase single-cell sequencing data via pseudobulk [15].

In some genomic regions, the the relative phase of *adjacent* bins may not be consistent. In particular, in regions where both haplotypes have similar average frequencies (i.e., $\frac{1}{P} \sum_p f_p(\boldsymbol{h}) \approx \frac{1}{P} \sum_p f_p(\overline{\boldsymbol{h}}) \approx 0.5$), the random variance in read counts between adjacent bins could produce inconsistent phases when looking across bins (Additional file 1: Fig. S8E, left). To address this issue, after mhBAF inference, we apply a heuristic on each chromosome arm to detect segments with a high density of haplotype switches, and in these segments, we phase haplotypes between bins (Additional file 1: Fig. S8E, right). See Additional file 1: Section S4 for details.

The mhBAF is distinguished from other approaches in the literature that summarize the ambiguity in the BAF signal caused by the unknown phasing of the two parental haplotypes. HATCHet [30] defines the *mirrored BAF* (mBAF) $\mathrm{mBAF}_p = \min\left(f_p(\boldsymbol{h}), 1 - f_p(\boldsymbol{h})\right)$ for each sample $p$. However, by taking the minimum separately in each sample $p$ rather than *across* samples, the mBAF does not necessarily measure the same haplotype in every sample (Additional file 1: Fig. S8A-C). As a result, the mBAF may fail to identify mirrored-subclonal copy-number events in which the more abundant allele is different across clones. Note that when only a single tumor sample is available, the mhBAF is equivalent to the mBAF.

The mhBAF is also a less biased estimator of allele frequencies than commonly used estimators for the mBAF and other related quantities. HATCHet [30] computes the mBAF by grouping together the smaller set of counts at each SNP position, i.e., by choosing $h_j$ to minimize the numerator in equation 1. We refer to this estimator as the *lower allele frequency* (LAF). In regions with equal proportions of both alleles (i.e., no allelic imbalance), the LAF is a biased estimator.

This is because the LAF chooses the smaller count at every locus resulting in an underestimate of the true mBAF. This bias is reduced by the EM approach in HATCHet2 to estimate the mhBAF. Other metrics that naively collapse parental haplotypes, including the squared log-odds ratio [31], suffer from similar biases near 0.5.

The mhBAF is also a more robust estimator of mirrored-subclonal CNAs, compared to earlier approaches [23, 39, 49, 50] that identify imbalanced segments that differ across samples in a post-hoc analysis. These methods are unable to detect regions of mirrored-subclonal allelic imbalance that are not identified by the input segmentation. Additionally, methods such as Battenberg [23] and MEDICC2 [50] that rely on the allelic imbalance in a single sample to identify haplotypes may fail to detect regions whose allelic imbalance differs across samples due to differences in clone mixture proportions (i.e., clone absences). In contrast, HATCHet2's EM algorithm pools information across all samples to infer the parental haplotypes and sample-specific mhBAF values.

### Clustering bins into copy-number states using local-global clustering

The next step in HATCHet2 is to group genomic bins into clusters, where each cluster corresponds to a distinct group of genomic bins that have the same haplotype-specific copy-number state in every clone. Existing CNA inference methods use two approaches to solve this clustering problem. The most widely used approach is to group *adjacent* bins along the genome into segments using piecewise constant fitting, hidden Markov models (HMMs), or other local approaches [22, 23, 25, 28, 29, 35, 36], relying on the assumption that a CNA affects multiple adjacent bins. However, these methods do not attempt to identify distant bins that share the same copy-number states during segmentation and typically do not group together bins across multiple samples, with a few exceptions [25, 67]. The second approach is to cluster all bins from all samples simultaneously, ignoring the genomic position of these bins. This *global* clustering approach was used in HATCHet [30] which was motivated by the application of simultaneous analysis of multiple samples and later extended in CHISEL [15] to cluster bins from thousands of individual cells[1]. However, because the global clustering in HATCHet and CHISEL does not leverage the expected contiguity of copy-number aberrations along each chromosome arm, the assignment of copy-number states to individual bins may be inaccurate, particularly for smaller focal aberrations.

HATCHet2 uses a *local-global clustering* approach based on an HMM that groups bins according to their RDR and mhBAF across all samples *as well as* their genomic positions. Specifically, let $(\mathbf{r_s}, \hat{\mathbf{f}}_s) = (r_{s,1}, \ldots, r_{s,P}, \hat{f}_{s,1}, \ldots, \hat{f}_{s,P})$ be the RDR and mhBAF values for bin $s$ across all $P$ tumor samples. We assume there are $K$ clusters with centroids

---

[1] A related approach employed by a few methods [31, 32] groups bins together across the genome independently in each sample.

$\mu_1, \ldots, \mu_K \in \mathbb{R}^{2P}$. Let $z_s \in [1, K]$ be the unobserved cluster assignment of bin *s*. We model this clustering problem using a hidden Markov model (HMM) with the following simple structure.

- There are *K* hidden states $1, \ldots, K$, each corresponding to a cluster.
- The transition matrix *A* containing the transition probabilities between hidden states has diagonal elements equal to $1 - \tau$ and off-diagonal elements equal to $\tau/(K - 1)$.
- The emissions are the RDR and mhBAF for each sample *s*: $< \mathbf{r_s} | \hat{\boldsymbol{f}_s} >$. The emission probabilities are multivariate Gaussian with mean $\mu_k$ and covariance $\Sigma_k$. By default, $\Sigma_k$ is diagonal (i.e., off-diagonal entries are constrained to be equal to 0).
- Local-global clustering is implemented using the Python class hmmlearn.hmm. GaussianHMM. The initial distribution is uniform and the prior distributions and their parameters are kept at their default settings.

We infer the parameters $\theta = \{\tau\} \cup \{\mu_k, \Sigma_k \text{ for all } k\}$ of this HMM using the Baum-Welch algorithm and infer the cluster assignment $z_s$ of each bin *s* using maximum *a posteriori* (MAP) estimation: $z_s = \text{argmax}_{k \in [1, K]} P(z_s = k | \boldsymbol{r}, \hat{\boldsymbol{f}}, \theta)$.

We assume that the cluster assignments of genomic bins from different chromosomes and different chromsome arms (i.e., p-arm vs. q-arm) are independent, the latter a reasonable assumption since centromeric reads are unaligned in current sequencing platforms. Thus, to fit multiple chromosome arms, we treat each arm as an independent string generated by the same model-i.e., all arms share the same parameters $\theta$ but their cluster assignments $z_s$ differ, and transitions between different chromosome arms are not considered. In practice, we perform model selection over *K* and choose the *K* that maximizes the silhouette score [103]. The architecture of this HMM is much simpler than other HMMs used for copy number inference, which have hidden states representing specific copy-number states (e.g., amplification, deletion, LOH, etc.) [24, 26, 29] and admixture by normal cells and multiple tumor clones [24]. The simple HMM used in HATCHet2 has fewer parameters to infer. In fact, it has only one additional parameter $\tau$ compared to the Gaussian mixture model (GMM) employed by HATCHet [30]; HATCHet's global clustering can be viewed as a "0th-order" HMM with Gaussian emissions.

To evaluate the local-global clustering, we applied it to 32 simulated datasets generated by MASCoTE from the original HATCHet publication [30] and evaluated how well each method recovered the true clustering of bins into copy-number states. We found that the Gaussian HMM used in HATCHet2 produced a more accurate clustering than the non-parametric GMM used in HATCHet or a parametric GMM using the same model selection approach as HATCHet2 (Additional file 1: Fig. S7). For this experiment, all three clustering methods were run on the data with fixed-width bins in the HATCHet publication repository [104].

### CHISEL single-cell haplotype-specific copy number

CHISEL [15] copy number calls for the 10x single-cell whole-genome sequencing data of a breast tumor were obtained from the CHISEL publication repository [105]. To compare these copy-number calls to those from HATCHet2, we first reconciled differences

between the two methods' derivation of (ordered) haplotype-specific copy numbers from (unordered) allele-specific copy-numbers. For example, in a region where both methods identify a clonal state with 3 copies of one allele and 1 copy of the other allele, HATCHet2 reports a haplotype-specific copy-number state of (3,1) while CHISEL might report a state of (1,3) across all cells. This distinction between the two haplotypes on a particular chromosome is arbitrary in the absence of additional data such as parental genotypes.

To reconcile CHISEL's haplotype assignments with those from HATCHet2, in regions of the genome where CHISEL did *not* identify mirrored-subclonal copy-number aberrations (e.g., some cells with state (2,1) and others with state (1,2)), we reassigned haplotypes according to HATCHet2's criterion: we defined the minor (*b*) haplotype as the haplotype that was less abundant on average across all cells in the sample. We defined mirrored-subclonal copy-number aberrations in the CHISEL results as segments of at least 5 consecutive bins in which at least 100 cells had allelic imbalance favoring one haplotype (i.e., state (*a*, *b*) where $a > b$) and at least 100 cells had allelic imbalance favoring the *other* haplotype (i.e., state (*a*, *b*) where $b > a$).

### HATCHet2 software improvements

HATCHet2 also includes several software enhancements that make HATCHet2 easier to integrate with existing bioinformatics workflows compared to HATCHet.

### Bioconda installation

HATCHet2 can be installed easily via Bioconda [51], an established ecosystem of bioinformatics software, and requires no manual building steps. Previously, HATCHet [30] had to be manually installed via downloading the source code and compiling a built-in C++ extension used to perform matrix factorization. New releases of HATCHet2 are frequently made on the HATCHet Bioconda page [106], allowing users to easily obtain the latest bug-fixes and features.

### Configurable steps

Individual steps in the HATCHet2 pipeline can be run *a la carte* using a single configuration file. Parameters of each processing step can be specified in this configuration file or given directly to the separate Python modules. This allows for cleaner customization and integration of the HATCHet2 pipeline in existing workflows.

### Resource requirements

The pre-binning steps of the HATCHet2 pipeline, which involve processing BAM files, took approximately 3–6 h using 24 processes on PowerEdge R730 machines with INTEL-XEON-E5-2680-V4 processors on datasets with up to 5 high-coverage (average depth 55X) WGS samples. The remaining steps can take between 5 and 10 h using 24 processes, again depending on the number of samples, depth of coverage, and complexity of the copy-number profiles.

### Support for ILP solvers

The last step of HATCHet2 computes integer-valued haplotype-specific copy numbers and clone proportions by solving an integer linear program (ILP). HATCHet2, like HATCHet [30], uses the Gurobi commercial optimizer [107] as the preferred engine for this step. Gurobi is fast and freely available for academic use. However, to broaden the utility of HATCHet2, we introduce support for a wide range of alternative ILP solvers. HATCHet2 is fully compatible with any solver that is supported by the Pyomo software package [108], including CPLEX [109] and the open-source GLPK [110]. This allows users to run HATCHet2 in environments that cannot or do not wish to use Gurobi due to licensing or deployment issues. By default, the ILP optimization in the HATCHet2 pipeline is run to optimality, and it was run to optimality for all results presented here. For the user's convenience, we expose parameters to limit the maximum runtime and memory of the ILP solver (see the documentation page  https://raphael-group.github.io/hatchet/doc_compute_cn.html  for additional details).

### Running HATCHet2 in the cloud

With the growing sizes of whole-genome sequencing datasets, cloud computing has become an attractive means of analyzing BAM files by circumventing the need to download these large files to local servers. Due to the software improvements mentioned above, running HATCHet2 on large, controlled-access datasets in any of the cloud computing platforms is now much easier. Once access to restricted datasets is obtained using federation services like eRA, minimal setup work on the part of the user (like installing the Google Cloud SDK) allows for the ability to run HATCHet2 entirely on the cloud. This mitigates security risks with data access and sidesteps local availability of computational resources. Using supplied helper scripts, HATCHet2 can be automated to run through massive genomic datasets with very little setup.

### Testing and documentation

The HATCHet2 code has been written to be readily testable across a variety of platforms, allowing us to use Continuous Integration tests in the HATCHet2 Github repository. These tests decrease turnaround time for new features and releases of HATCHet2 and encourage community contributions.

We also use Sphinx (www.sphinx-doc.org/) to automatically generate updated documentation, which provides a comprehensive guide to the installation of HATCHet2 and the usage of steps in the HATCHet2 pipeline.

### Benchmarking on simulated data

We compared HATCHet2 to HATCHet [30], cloneHD [25], TITAN [24], and Battenberg [23] using 8 multi-sample simulated datasets generated by the MASCoTe simulator [30]. The datasets evaluated here are a subset of the datasets originally analyzed in the HATCHet publication [30]. Thus, we report the results for cloneHD, TITAN, and Battenberg on these datasets from the HATCHet publication (available in the HATCHet publication repository [104]). To ensure that the parameter settings for

HATCHet were as similar as possible to those of HATCHet2, we reran HATCHet on the multi-sample datasets. Additionally, to assess the single-sample performance of HATCHet2 and HATCHet, we applied both methods to each of the 32 tumor samples independently. We used the same parameter settings for each method on all multi-sample datasets, and a second set of parameter settings for each method on all single-sample datasets. The full parameter specifications are listed in the `hatchet.ini` files for each dataset in the HATCHet2 publication repository [111].

See Additional file 1: Section S1.2 for details on the metrics used to evaluate performance on simulated data.

## Supplementary Information

> Additional file 1. Supplement. Supplemental methods sections S1-S10 and supplemental figures S1-S14.
>
> Additional file 2. Review history.

**Peer review information**

**Review history**
The review history is available as Additional file 2.

**Authors' contributions**
B.J.R., M.A.M., and S.Z. initiated the project; M.A.M. analyzed the data; M.A.M., V.B., B.J.A, M.B., and S.Z. developed the new software used in the work; V.B., M.A.M., B.J.A., M.B., and K.M. tested the software; M.A.M., B.J.R., S.Z., and B.J.A. wrote and revised the manuscript; B.J.R. and S.Z. supervised the project.

**Funding**
This research is supported by NIH/NCI grants U24CA248453 and U24CA264027 awarded to B.J.R. B.J.A gratefully acknowledges financial support from the Schmidt DataX Fund at Princeton University, made possible through a major gift from the Schmidt Futures Foundation.

**Availability of data and materials**
The source code for HATCHet2 is available in the HATCHet2 GitHub repository [112] and conda-installable via the HATCHet2 bioconda recipe [106]. The simulated datasets used for benchmarking in the main text of this paper as well as the TITAN [24], cloneHD [25], and Battenberg [23] results on these data are available in the HATCHet publication repository [104], which also contains the HATCHet results on real data. The real datasets used in this paper are available with their respective original publications: prostate cancer dataset [42] and single-cell breast cancer dataset [54]. The HATCHet2 results on both simulated and real data, the HATCHet results on simulated data, and the source code and data for the additional purity and mirrored-subclonal benchmarking simulated datasets are available in the HATCHet2 publication repository [111]. The version of the HATCHet2 source code used to generate the results reported in this manuscript and the supplement is available in Zenodo [113].

## Declarations

**Ethics approval and consent to participate**
Not applicable.

**Consent for publication**
Not applicable.

**Competing interests**
The authors declare no competing interests.

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

## 