## [Additional file 2. Review history. · Genome Biology]

Review History

First round of review

Reviewer 1

Were you able to assess all statistics in the manuscript, including the appropriateness of statistical tests used? No.

Were you able to directly test the methods? Yes.

Comments to author:

In Myers et al., the authors present their allelic-specific copy number algorithm, HATCHet2 which is an improvement on their previous algorithm HATCHet. The main innovation with HATCHet2 is that it phases SNPs and generates haplotype-specific copy number. HATCHet2 also takes advantage of variable bin sizes to more evenly distribute heterozygous SNPs and cluster bins across samples for better identification of focal copy number alterations. Using simulated WGS copy number data, the authors demonstrate that HATCHet2 is more accurate than other algorithms. The authors have made the installation of HATCHet2 easier.

This algorithm will be of interest to those who analyze cancer WGS data, especially researchers who study tumor clonality and evolution. Before this manuscript gets accepted for publication, the authors should make some clarifications on the methods and analysis modifications.

Specific points.

1. Most allelic copy number algorithms require some sort of manual intervention to ensure that the modeled integer copy number states best fit the allelic data. This is especially true in deciding if WGD events have occurred. It's unclear from the manuscript if HATCHet2 requires a manual review step or if the best scoring solution is just selected. If a manual review is not done, could the accuracy of HATCHet2 in simulations be further improved by doing this?
2. In supplement S1, the denominator of the formula for computing accuracy is $|Ss \cup Ss|$, but it is not clear precisely what this is. Is this the union based on sequences covered by segments and not CN states?
3. Hyper-segmentation is an issue that plagues many copy number callers. While the increase in mean segment size suggests that fewer segments are being generated, the number of segments should also be included for each sample/method in the violin plots shown in Figure 2a.
4. The performance of allelic copy number algorithms breaks down in tumors of low purity. Understanding these limits of HATCHet2 would be especially important for those who want to use it with cell-free DNA. It would be helpful for potential users, if the accuracy of HATCHet2 could be benchmarked in simulated copy number data of decreasing amounts of tumor purity in order to quantify the limits of the algorithm.
5. HATCHet2 clusters bins across samples as part of a step to produce locality-aware clustering. Does this suggest that HATCHet2 requires samples of the same tumor type or subtype that have similar, albeit not identical, copy number profiles in order to achieve the best performance? What

would happen to the performance of HATCHet2 if it were run on a cohort consisting of patients of different molecular ancestry with diverse haplotypes?

6. The genes selected to be shown in Figures 2 and 3 come from the COSMIC Cancer Gene Census for prostate cancer, but this set is a little odd. It includes CANT1, a rare fusion partner for ETV4, which has only been reported in single patients in two PubMed manuscripts published 15 years ago. Surprisingly the COSMIC Cancer Gene Consensus misses well-known genes that are frequently altered by copy number in prostate cancer (please see The Cancer Genome Atlas Research Network, Cell 2014). Notable absent genes affected by focal copy number alterations include MYC, RB1, and CDH1. This analysis should be updated to include genes that are frequently altered by copy number in prostate cancer.

Reviewer 2

Were you able to assess all statistics in the manuscript, including the appropriateness of statistical tests used? Yes: I believe I could appropriately assess all statistical tests and saw no problems or need for further assessment.

Were you able to directly test the methods? No.

Comments to author:

This paper presents HATCHet2, a new computer program for identifying clone- and haplotype-specific copy numbers from tumor genomic samples. The work addresses an important problem in analyzing tumor genomic data. Inferences of CNAs are of independent value in understanding tumor evolution and tools for this task are widely used as preprocessing steps in reconstructing clonal lineage trees due to the difficulty of interpreting SNV data when CNAs are unknown. The work offers some innovative ideas for improving on the prior methods, notably a new statistical test used in estimating copy numbers and an improved strategy for binning and clustering copy number segments across samples. The software appears to be very well designed, with good attention to portability and ease of use. The validation testing is thorough and makes a convincing case that HATCHet2 is on balance an improvement over the prior art for this task. I expect the method to be a valuable and welcome addition to bioinformatics pipelines for cancer genome analysis. Given this, my questions and criticisms are relatively few and minor:

1. The paper does a very nice job of considering combinations of simulated data, bulk data, and pseudobulk data derived from single cell sequence when validating. However, I would have appreciated seeing a bit more from the simulated data about the conditions in which HATCHet2 is particularly effective or ineffective compared to prior methods. Does it do better with tumors that have large numbers of CNAs or fewer? Is it better at tumors with large CNAs, small CNAs, or a wide range? Does it deal well with highly amplified loci? Does it matter how we assume the CNAs arose, such as short tandem duplications, chromosome or chromosome-arm scale gains or losses, or extra chromosomal DNA? While I realize there is an endless list of questions like this one might ask, I think a few basic insights would be useful to users and potentially for future improvements of methods in this space.

2. It is nice to see the attention to haplotype phasing and in particular allowing correction for rare haplotypes relative to using a purely reference-based method. I think the field in general is

becoming more sensitive to whether methods are likely to underperform for groups poorly represented in reference datasets. I am not sure I fully followed how this works, though. Is the phasing algorithm based purely on examining read depths of each SNP in isolation or is it making any use of which SNPs co-occur on single sequence reads? That is, is it using a "haplotype assembly" style approach to reject haplotypes inconsistent with individual reads or just looking for discrepancies based on single-SNP read depths?

3. Unless I have missed it, there was no comment on run times or hardware needs for the software. I think it would be useful to have some idea of the general needs. Will it run efficiently on standard desktop hardware or do you need a cloud/cluster for typical datasets? Some comparison to the alternative methods considered here would also be useful. I do not see anyone caring very much if they are all about the same order of magnitude or if they all run quickly enough not to be a bottleneck in anyone's pipeline, but large differences could be problematic.

4. Related to the prior point, the final phase of the algorithm relies on integer linear programming (ILP), which may take exponential run time in worst cases. Are run times for this step always practical and does the ILP solver always run to optimality? Is there some way this is guaranteed or some fallback if a user hits a hard case that will not terminate in a reasonable time?

Reviewer 3

Were you able to assess all statistics in the manuscript, including the appropriateness of statistical tests used? No.

Were you able to directly test the methods? No.

Comments to author:

The authors present HATCHet2, a method for inferring haplotype- and clone-specific CNAs from single or multiple bulk WGS tumours. HATCHet2 builds on HATCHet (Zaccaria et al. 2020). The updated method differs from its predecessor in the following ways:

1) It incorporates phasing of heterozygous SNPs from the matched normal for improved BAF estimates per bin

2) It uses a greedy algorithm to bin the reference into variable-width bins in order to ensure a similar number of SNPs within each bin with the goal of reducing variance in BAF

3) It introduces a modified measure of allelic imbalance called mhBAF. For every bin, SNPs are phased and assigned to a haplotype. The haplotype with lower average abundance across multiple samples is identified and considered the minor or B-allele. For each sample, the frequency of this B-allele is then computed. This is in contrast to previous methods, which identified the minor allele on a per-sample basis. The goal here is to improve identification of mirrored subclonal allelic imbalance across samples.

4) It identifies bins with similar read-depth ratios (RDR) and mhBAF using an HMM, in contrast with the global clustering that ignores genomic location.

The fifth and final step, assigning copy number and clone proportions, is performed using the same matrix factorization algorithm used in HATCHet.

The authors apply HATCHet2 to simulated and real datasets. When applied to simulated data, results suggest similar performance to HATCHet in the multi-sample context and slightly

improved performance in the single-sample context. Application to 50 prostate cancer samples from 10 patients reveals mirrored subclonal CNAs, and the authors suggest identification of focal amplifications is improved. Finally, application to pseudobulk data from multi-sample single cell WGS shows HATCHet2 can detect two dominant clones and estimates similar clone proportions.

The paper is well written, modifications from HATCHet are generally clearly described, and the choice of datasets appropriate. Some additional benchmarking and methodological details are needed.

Major comments

1. One of the main claims is that HATCHet2 has improved performance for focal CNAs, but no benchmarking is shown to support this. Figure S1 shows accuracy for five tools on the full simulated dataset - a similar figure focused on simulated focal amplifications should be included.

For the prostate dataset, HATCHet2 has on average longer segments than HATCHet and shorter than Battenberg, and a couple of examples of over/under segmentation by these tools is shown. To support the claim the authors could present, for example, the number of focal amps called by each of these tools in the prostate cohort, how many of these are shared between tools, and what proportion overlap known oncogenes.

2. By introducing mhBAF, another main aim of the method is to improve detection of mirrored subclonal CNAs. Again examples are shown but no rigorous comparison of accuracy. Can a simulated dataset be generated with mirrored subclonal CNAs for benchmarking? How many such events reported by HATCHet2 in the prostate dataset are missed by HATCHet and Battenberg?

3. How accurate is HATCHet2 when it comes to inferring the number of clones and their proportions on the simulated datasets?

4. Figure 3D how does "clone 1" differ from "clone N" and how does "clone 3" differ from "clone 4"? Are they truly distinct clones with differences in other regions of the genome but not the region of interest for this figure, or is the model inferring multiple similar clones? Including clone profiles similar to Figure 3 B,C for the other clones and highlighting differences between them would be helpful. Similarly, results for other patients from the prostate dataset can be shown in the supplement.

5. Additional methodological details are needed, including: details of the E- and M-steps for the mhBAF estimation, does the HMM model include priors, what parameter/hyperparameter settings were used for the analyses throughout?

6. How were user-tunable parameters were selected, and how does modifying them impact performance (minimum number of SNPs/total reads per bin, maximum CN, minimum clone frequency, etc.)?

7. In Figure 4, it looks like focal amplifications identified by HATCHet2 and CHISEL are quite different. Can the authors explain why this is?

Minor comments

1. Naming the modified BAF "mirrored haplotype B-allele frequency" is confusing in that it implies this statistic solely an indicator of mirrored CNAs. The term used in the caption of Figure S6 "minor haplotype BAF" seems (to this reviewer) more appropriate and easily understood.

2. Similar to the above, calling the HMM "locality aware clustering" in the introduction is unnecessarily confusing. HMMs are common in CNA inference and most readers will likely be familiar with them. The authors do make clear that the simple Gaussian HMM isn't used to infer integer copy number but only to assign latent states to bins with similar distributions and genomic proximity. That explanation is fine - no need to give it a new term.

3. Many CNA callers have a hyperparameter that can be modified to reduce over/under segmentation. How were parameters set for the competing methods?

4. Can HATCHet2 detect subclonal WGD or does it assume ploidy is clonal? The supplement indicates that in the simulated datasets any WGD affected all clones.

5. For Figure 2, S2, and S3, the RDR and mhBAF values appear identical for all tools. Is this because all of the tools were run with the same input as HATCHet2, or were the inferred CN calls for those tools simply mapped onto the HATCHet2 RDR and mhBAF values for visualization? This should be made clear.

6. Figure 2 C,D,E,F, Figure 3 B,C, Figure S2 and S3 need a colour legend.

7. Figure 3 B and C have "fractional copy number" on the y-axis. This isn't defined.

8. Typo on page 14 "individual cells".

We thank the reviews for the constructive feedback on our manuscript. Please see below our point-by-point responses (in blue text) to the comments of each reviewer (black text).

Reviewer #1

In Myers et al., the authors present their allelic-specific copy number algorithm, HATCHet2 which is an improvement on their previous algorithm HATCHet. The main innovation with HATCHet2 is that it phases SNPs and generates haplotype-specific copy number. HATCHet2 also takes advantage of variable bin sizes to more evenly distribute heterozygous SNPs and cluster bins across samples for better identification of focal copy number alterations. Using simulated WGS copy number data, the authors demonstrate that HATCHet2 is more accurate than other algorithms. The authors have made the installation of HATCHet2 easier.

This algorithm will be of interest to those who analyze cancer WGS data, especially researchers who study tumor clonality and evolution. Before this manuscript gets accepted for publication, the authors should make some clarifications on the methods and analysis modifications.

Specific points.

1. Most allelic copy number algorithms require some sort of manual intervention to ensure that the modeled integer copy number states best fit the allelic data. This is especially true in deciding if WGD events have occurred. It's unclear from the manuscript if HATCHet2 requires a manual review step or if the best scoring solution is just selected. If a manual review is not done, could the accuracy of HATCHet2 in simulations be further improved by doing this?

HATCHet2 does not require a manual review step and will automatically select the number of clones and the WGD status that best fits the data. The default settings of HATCHet2 provide good performance on most typical datasets (e.g. WGS with coverage at least 30X, WES with coverage at least 100X). However, as the reviewer notes HATCHet2 – like other allelic copy number algorithms – can benefit from manual review as the quality of results can depend on tumor purity and the structure of the tumor genome. On the simulated datasets in the manuscript, the user-tunable parameters were set once for all instances, and HATCHet2 automatically selected other parameters including the number of clusters and the WGD status. On real datasets, the parameters were manually tuned to optimize results. The detailed parameter settings used for every dataset is included in the publication repository: <https://github.com/raphael-group/hatchet2-paper>

It is likely that the results on simulated data could be further improved by manually tuning parameters for each instance, but our intent was to demonstrate the performance of HATCHet2 without expert parameter tuning.

The documentation for HATCHet2 includes descriptions of all parameters for each step (https://raphael-group.github.io/hatchet/doc_fullpipeline.html#detailed-steps), as well as detailed recommendations on parameter tuning for the steps most sensitive to parameter settings.

1. Binning: increasing the minimum number of SNP/total reads per bin produces larger bins that tend to have a cleaner signal at the cost of fitting small events more poorly, as we describe in the “Analyze different type of data” recommendation page: https://raphael-group.github.io/hatchet/recommendation_datatype.html
2. Clustering: by default HATCHet2 explores a range of values for the number K of clusters, but users may want to fine-tune the number of clusters. We provide guidance in the “Analyze global clustering” recommendation page: https://raphael-group.github.io/hatchet/recommendation_clustering.html
 - a. In addition to the plots produced by HATCHet2, a recent tool CNAViz (Lalani et al., *PLOS Comput. Biol.* 2022) can be used to interactively inspect and edit the clustering solution produced by HATCHet2: <https://github.com/elkebir-group/cnaviz>
3. Inference: by default HATCHet2 automatically selects the clonal cluster, infers WGD status, and chooses the correct number of tumor clones (testing 1-7 tumor clones by default). Users may want to fine-tune these parameters and others, and we provide guidance on doing so in the “Analyze HATCHet’s inference” recommendation page: https://raphael-group.github.io/hatchet/recommendation_inference.html

The HATCHet2 pipeline automatically produces plots after the clustering step that can be used to troubleshoot binning and clustering, and after the inference step that can be used to troubleshoot the factorization step. Each of these steps can be run and tuned independently or jointly, as the HATCHet2 pipeline is highly modular. The parameters used for each step of HATCHet2, as well as the steps to be run, can be specified a single .ini file (referred to in the documentation as “hatchet.ini”) and run using a single command:

https://github.com/raphael-group/hatchet/blob/master/docs/source/doc_runhatchet.md

We describe the parameter settings used on simulated data in the text in the Methods section “Benchmarking on simulated data”:

“We compared HATCHet2 to HATCHet \cite{hatchet}, cloneHD \cite{clonehd}, TITAN, and Battenberg using 8 multi-sample simulated datasets generated by the MASCoTe simulator. The datasets evaluated here are a subset of the datasets originally analyzed in the HATCHet publication. Thus, we report the results for cloneHD, TITAN, and Battenberg on these datasets from the HATCHet publication (available at <https://github.com/raphael-group/hatchet-paper>). To ensure that the parameter settings for HATCHet were as similar as possible to those of HATCHet2, we reran HATCHet on the multi-sample datasets.

Additionally, to assess the single-sample performance of HATCHet2 and HATCHet, we applied both methods to each of the 32 tumor samples independently. We used the same parameter settings for each method on all multi-sample datasets, and a second set of parameter settings for each method on all single-sample datasets. The full parameter specifications are listed in the hatchet.ini files for each dataset in the HATCHet2 publication repository: <https://github.com/raphael-group/hatchet2-paper>.”

2. In supplement S1, the denominator of the formula for computing accuracy is $|S_s \cup T_s|$, but it is not clear precisely what this is. Is this the union based on sequences covered by segments and not CN states?

S_s represents the set of inferred copy-number states (where a copy-number state is a pair (a,b) of integers) for region s , and T_s represents the set of true copy-number states for region s . The denominator for accuracy is the total number of states either in the ground truth solution or inferred by the method, and the contribution of each region s to the metric is weighted by its size. We updated the text of the supplement to clarify this:

“Let S_s indicate the set of haplotype-specific copy-number states that a method infers for a particular genomic segment s across the n tumor clones...Let T_s indicate the set of true haplotype-specific copy-number states for region s .”

3. Hyper-segmentation is an issue that plagues many copy number callers. While the increase in mean segment size suggests that fewer segments are being generated, the number of segments should also be included for each sample/method in the violin plots shown in Figure 2a.

We added a track to Figure 2A, reproduced here, that indicates the number of segments inferred by each method:

4. The performance of allelic copy number algorithms breaks down in tumors of low purity. Understanding these limits of HATCHet2 would be especially important for those who want to use it with cell-free DNA. It would be helpful for potential users, if the accuracy of HATCHet2 could be benchmarked in simulated copy number data of decreasing amounts of tumor purity in order to quantify the limits of the algorithm.

To evaluate the sensitivity of HATCHet2 to detect low-purity tumor clones, we tested HATCHet2 on 191 simulated datasets – each dataset consisting of 2-3 tumor clones and 2

tumor samples. In each dataset, one of the two samples was a high-purity tumor sample (purity 0.9) and the other was an admixed tumor sample with varying (lower) purity. This design was meant to emulate a research setting in which an initial high-purity sample was sequenced along with one (or potentially more) cell-free DNA samples or liquid biopsies at lower purity. In the admixed sample, all clones were present at equal proportions ranging from 0.05 to 0.3. We found that while HATCHet2 performed poorly on the samples with clone proportions at 0.05, it achieves reasonable accuracy on the remaining simulations (with some decrease in performance at the upper end of the tested range, likely due to ambiguity between the different clones present in the “liquid” sample). We reproduce the supplemental figure and its caption below.

Evaluation of HATCHet2 on simulated datasets with varying purity. Results on 191 simulated 2-sample datasets in which one “liquid” sample was of lower purity. Proportions of each clone ($n \in \{2,3\}$) in the liquid sample was equal (x-axis). Performance was evaluated using AASAPGP (y-axis).

5. HATCHet2 clusters bins across samples as part of a step to produce locality-aware clustering. Does this suggest that HATCHet2 requires samples of the same tumor type or subtype that have similar, albeit not identical, copy number profiles in order to achieve the best performance? What would happen to the performance of HATCHet2 if it were run on a cohort consisting of patients of different molecular ancestry with diverse haplotypes?

HATCHet2 is designed to be run on one or more tumor samples that share the *same* evolutionary origin, i.e., from the same patient, rather than a cohort of different individuals. Thus, this issue of a highly heterogeneous cohort does not arise. We clarified this in the text with the following:

> *"Here we introduce HATCHet2 (Holistic Allele-specific Tumor Copy-number Heterogeneity 2), a method to infer clone-specific and haplotype-specific CNAs from DNA sequencing data from one or more bulk tumor samples from the same patient."*

We also added similar context to other places in the text in which the input to HATCHet2 is described, including the caption to Fig. 1, the beginning of the Results section, and the Discussion section,

6. The genes selected to be shown in Figures 2 and 3 come from the COSMIC Cancer Gene Census for prostate cancer, but this set is a little odd. It includes CANT1, a rare fusion partner for ETV4, which has only been reported in single patients in two PubMed manuscripts published 15 years ago. Surprisingly the COSMIC Cancer Gene Consensus misses well-known genes that are frequently altered by copy number in prostate cancer (please see The Cancer Genome Atlas Research Network, Cell 2014). Notable absent genes affected by focal copy number alterations include MYC, RB1, and CDH1. This analysis should be updated to include genes that are frequently altered by copy number in prostate cancer.

We thank the reviewer for the correction and we agree that the gene list from the 2015 TCGA prostate cancer paper is more relevant. We thus updated Figures 2-3 to use the TCGA gene list instead, and moved the panels with the CGC gene list to the supplement. The new panels and text are reproduced here:

Number of segments within 1 MB of gene

“We focused on regions near 41 genes highlighted in The Cancer Genome Atlas (TCGA) prostate cancer publication (TCGA network, Cell 2015). We counted the number of copy-number segments reported within 1 megabase of each gene (Fig. 2B). Consistent with the results for the length distribution of segments, we observed that Battenberg yields only a single segment spanning the entire region for 367/410 gene-patient pairs (89.5%). In contrast, HATCHet often has many segments surrounding these genes: up to 21 segments within 1 Mb from ATM in patient A29. HATCHet2 infers fewer segments than HATCHet, producing simpler copy-number profiles, but infers more segments than Battenberg, which can better fit the data when there is evidence of copy-number change. To demonstrate the advantages of HATCHet2 and their impact, we focus in detail on two examples involving genes with a key role in prostate cancer: TP53 and CANT1.”

Reviewer #2

This paper presents HATCHet2, a new computer program for identifying clone- and haplotype-specific copy numbers from tumor genomic samples. The work addresses an important problem in analyzing tumor genomic data. Inferences of CNAs are of independent value in understanding tumor evolution and tools for this task are widely used as preprocessing steps in reconstructing clonal lineage trees due to the difficulty of interpreting SNV data when CNAs are unknown. The work offers some innovative ideas for improving on the prior methods, notably a new statistical test used in estimating copy numbers and an improved strategy for binning and clustering copy number segments across samples. The software appears to be very well designed, with good attention to portability and ease of use. The validation testing is thorough and makes a convincing case that HATCHet2 is on balance an improvement over the prior art for this task. I expect the method to be a valuable and welcome addition to bioinformatics pipelines for cancer genome analysis. Given this, my questions and criticisms are relatively few and minor:

1. The paper does a very nice job of considering combinations of simulated data, bulk data, and pseudobulk data derived from single cell sequence when validating. However, I would have appreciated seeing a bit more from the simulated data about the conditions in which HATCHet2 is particularly effective or ineffective compared to prior methods. Does it do better with tumors that have large numbers of CNAs or fewer? Is it better at tumors with large CNAs, small CNAs, or a wide range? Does it deal well with highly amplified loci? Does it matter how we assume the CNAs arose, such as short tandem duplications, chromosome or chromosome-arm scale gains or losses, or extra chromosomal DNA? While I realize there is an endless list of questions like this one might ask, I think a few basic insights would be useful to users and potentially for future improvements of methods in this space.

To address this, we added a section “Performance on simulated data as a function of segment size” to the supplement and Figure S9 to the supplement which investigate performance by segment size on the simulated data. We reproduce the section, figure, and its caption below.

“To demonstrate the performance of HATCHet2 in inferring copy-number events of varying sizes, we evaluated the recall and precision of HATCHet2, HATCHet, cloneHD, TITAN, and Battenberg on the MASCoTE simulated multi-sample datasets described in Results. HATCHet2 and HATCHet outperformed the best of the other 3 methods across size ranges on both methods (average improvement in precision by 133% and recall by 78% vs. the third-best method for each size range and dataset), while the HATCHet2 and HATCHet performed comparably (improvement in precision by 1.7% and decrease in recall by 0.006%; Fig. S2A-B). We then compared HATCHet2 to HATCHet on single-sample datasets by considering each MASCoTE simulated sample independently, and found that HATCHet2 outperformed HATCHet by a larger margin (6.2% higher precision, 5.7% higher recall; Fig. S2C-D). Moreover, on segments smaller than 1 Mb, HATCHet2 achieved 16.6% higher precision and 15.2% higher recall than HATCHet.”

Benchmarking HATCHet2, HATCHet, TITAN, cloneHD, and Battenberg on simulated data by event size. **A-B** Results from all methods on 8 multi-sample datasets simulated using MASCoTE measured by recall (A) and precision (B), stratified by the size of the ground truth simulated segment (x-axis). **C-D** Results on the same data with each sample run independently (n=32 samples) measured by recall (C) and precision (D), stratified by the size of the ground truth simulated segment (x-axis).

2. It is nice to see the attention to haplotype phasing and in particular allowing correction for rare haplotypes relative to using a purely reference-based method. I think the field in general is becoming more sensitive to whether methods are likely to underperform for groups poorly represented in reference datasets. I am not sure I fully followed how this works, though. Is the phasing algorithm based purely on examining read depths of each SNP in isolation or is it making any use of which SNPs co-occur on single sequence reads? That is, is it using a "haplotype assembly" style approach to reject haplotypes inconsistent with individual reads or just looking for discrepancies based on single-SNP read depths?

The phasing algorithm used by default in the HATCHet2 pipeline (SHAPEIT2, O'Connel et al., PLoS Genet. 2014) does not make any use of SNPs which co-occur on individual sequence reads. Instead, given the homozygous/heterozygous status of SNPs called in the patient, SHAPEIT2 evaluates which of these SNPs are likely to occur on the same haplotype using a large reference database of known human haplotypes. Since this procedure depends on a reference haplotype database, it may perform poorly for individuals from populations that are poorly represented in the reference haplotype database. We now note this limitation in the revised Methods section:

Reference-based phasing gives only an approximation of the haplotype phase, and

switch errors -- erroneously switching phase between parental haplotypes -- occur with appreciable rates (Choi et al, PLOS Genetics 2018). These errors may be due to errors in the reconstructed haplotypes within the reference panel (particularly for rare alleles (Belsare et al, BMC Genomics 2019) or because the target individual may have a novel haplotype not present in the panel. Indeed, reference-based phasing may be less accurate for individuals from populations that are poorly represented in the reference database (De Marino et al., PLOS ONE 2022).

HATCHet2 includes a haplotype switch error correction step to correct for potential errors in the reference-based phasing (described in Supplement section “Haplotype switch correction algorithm”), which can help to address this issue.

As an alternative, a HATCHet2 user could run their preferred phasing method or submit their data to an imputation server (e.g., <https://imputation.biodatacatalyst.nhlbi.nih.gov/> or <https://imputationserver.sph.umich.edu/index.html#!>) and supply the resulting phased VCF file as input to the binning step of the HATCHet2 pipeline:

https://raphael-group.github.io/hatchet/doc_combine_counts.html

3. Unless I have missed it, there was no comment on run times or hardware needs for the software. I think it would be useful to have some idea of the general needs. Will it run efficiently on standard desktop hardware or do you need a cloud/cluster for typical datasets? Some comparison to the alternative methods considered here would also be useful. I do not see anyone caring very much if they are all about the same order of magnitude or if they all run quickly enough not to be a bottleneck in anyone’s pipeline, but large differences could be problematic.

HATCHet2 can be run on standard desktop hardware, although we would recommend using a computing cluster or cloud resources when possible. We did not run TITAN, Battenberg, or cloneHD for this manuscript, but we have included here some figures showing the runtime of HATCHet and HATCHet2 on the Gundem BAM files up to the binning step (left) and on simulated read counts from the binning step onward (right):

Additionally, to clarify the resource requirements of HATCHet2 to the reader, we added the following text to the Methods section:

“The pre-binning steps of the HATCHet2 pipeline, which involve processing BAM files, took approximately 3-6 hours using 24 processes on PowerEdge R730 machines with INTEL-XEON-E5-2680-V4 processors on datasets with up to 5 high-coverage (average depth 55X) WGS samples. The remaining steps can take between 5 and 10 hours using 24 processes, again depending on the number of samples, depth of coverage, and complexity of the copy-number profiles.”

The HATCHet2 pipeline is highly modular and could be started or restarted at various steps, including to test parameter settings or recover from an aborted job (details in the documentation: https://raphael-group.github.io/hatchet/doc_fullpipeline.html#detailedsteps).

4. Related to the prior point, the final phase of the algorithm relies on integer linear programming (ILP), which may take exponential run time in worst cases. Are run times for this step always practical and does the ILP solver always run to optimality? Is there some way this is guaranteed or some fallback if a user hits a hard case that will not terminate in a reasonable time?

The factorization problem in HATCHet2 is solved using a coordinate descent algorithm as in HATCHet (Zaccaria and Raphael, 2022) in which each coordinate step is a distinct ILP. The ILPs may not be always be solved to optimality, but it is initialized with a heuristic feasible solution and users can specify a maximum time for this step of the pipeline:

https://raphael-group.github.io/hatchet/doc_compute_cn.html

By default, this parameter is not set, and all results in the paper were solved to optimality. We added the following text to the Methods section to clarify this to the reader.

By default, the ILP optimization in the HATCHet2 pipeline is run to optimality, and it was run to optimality for all results presented here. For the user's convenience, we expose parameters to limit the maximum runtime and memory of the ILP solver (see https://raphael-group.github.io/hatchet/doc_compute_cn.html for additional details).

Reviewer #3

The authors present HATCHet2, a method for inferring haplotype- and clone-specific CNAs from single or multiple bulk WGS tumours. HATCHet2 builds on HATCHet (Zaccaria et al. 2020). The updated method differs from its predecessor in the following ways:

1) It incorporates phasing of heterozygous SNPs from the matched normal for improved BAF estimates per bin

2) It uses a greedy algorithm to bin the reference into variable-width bins in order to ensure a similar number of SNPs within each bin with the goal of reducing variance in BAF

3) It introduces a modified measure of allelic imbalance called mhBAF. For every bin, SNPs are phased and assigned to a haplotype. The haplotype with lower average abundance across multiple samples is identified and considered the minor or B-allele. For each sample, the

frequency of this B-allele is then computed. This is in contrast to previous methods, which identified the minor allele on a per-sample basis. The goal here is to improve identification of mirrored subclonal allelic imbalance across samples.

4) It identifies bins with similar read-depth ratios (RDR) and mhBAF using an HMM, in contrast with the global clustering that ignores genomic location.

The fifth and final step, assigning copy number and clone proportions, is performed using the same matrix factorization algorithm used in HATCHet.

The authors apply HATCHet2 to simulated and real datasets. When applied to simulated data, results suggest similar performance to HATCHet in the multi-sample context and slightly improved performance in the single-sample context. Application to 50 prostate cancer samples from 10 patients reveals mirrored subclonal CNAs, and the authors suggest identification of focal amplifications is improved. Finally, application to pseudobulk data from multi-sample single cell WGS shows HATCHet2 can detect two dominant clones and estimates similar clone proportions.

The paper is well written, modifications from HATCHet are generally clearly described, and the choice of datasets appropriate. Some additional benchmarking and methodological details are needed.

Major comments

1. One of the main claims is that HATCHet2 has improved performance for focal CNAs, but no benchmarking is shown to support this. Figure S1 shows accuracy for five tools on the full simulated dataset - a similar figure focused on simulated focal amplifications should be included.

To address this, we added a section "Performance on simulated data as a function of segment size" to the supplement and Figure S9 to the supplement which investigate performance by segment size on the simulated data. We reproduce the section, figure, and its caption below.

"To demonstrate the performance of HATCHet2 in inferring copy-number events of varying sizes, we evaluated the recall and precision of HATCHet2, HATCHet, cloneHD, TITAN, and Battenberg on the MASCoTE simulated multi-sample datasets described in Results. HATCHet2 and HATCHet outperformed the best of the other 3 methods across size ranges on both methods (average improvement in precision by 133% and recall by 78% vs. the third-best method for each size range and dataset), while the HATCHet2 and HATCHet performed comparably (improvement in precision by 1.7% and decrease in recall by 0.006%; Fig. S2A-B). We then compared HATCHet2 to HATCHet on single-sample datasets by considering each MASCoTE simulated sample independently, and found that HATCHet2 outperformed HATCHet by a larger margin (6.2% higher precision, 5.7% higher recall; Fig. S2C-D). Moreover, on segments smaller than 1 Mb, HATCHet2 achieved 16.6% higher precision and 15.2% higher recall than HATCHet."

Benchmarking HATCHet2, HATCHet, TITAN, cloneHD, and Battenberg on simulated data by event size. **A-B** Results from all methods on 8 multi-sample datasets simulated using MASCoTE measured by recall (A) and precision (B), stratified by the size of the ground truth simulated segment (x-axis). **C-D** Results on the same data with each sample run independently (n=32 samples) measured by recall (C) and precision (D), stratified by the size of the ground truth simulated segment (x-axis).

For the prostate dataset, HATCHet2 has on average longer segments than HATCHet and shorter than Battenberg, and a couple of examples of over/under segmentation by these tools is shown. To support the claim the authors could present, for example, the number of focal amps called by each of these tools in the prostate cohort, how many of these are shared between tools, and what proportion overlap known oncogenes.

To address this, we added a section “Focal amplifications identified by HATCHet2, HATCHet, and Battenberg in prostate cancer” and Figure S11 to the supplement which investigate the focal amplifications called by each method on the Gundem prostate cancer dataset and the overlap between said amplifications with TCGA prostate cancer genes. We reproduce the figure and its caption below.

Focal amplifications identified by HATCHet2, HATCHet, and Battenberg on prostate cancer patients. A Number of focal amplifications – defined as segments shorter than 1 Mb with total copy number 4 or higher in at least 1 clone – reported by HATCHet2, HATCHet, and Battenberg on 10 prostate cancer patients from Gundem et al. (*Nature* 2015). **B** Breakdown of focal amplifications called by HATCHet2 in terms of overlap with those called by other methods. **C** Number of focal amplifications called by each method that overlap TCGA prostate cancer genes.

2. By introducing mhBAF, another main aim of the method is to improve detection of mirrored subclonal CNAs. Again examples are shown but no rigorous comparison of accuracy. Can a simulated dataset be generated with mirrored subclonal CNAs for benchmarking? How many such events reported by HATCHet2 in the prostate dataset are missed by HATCHet and Battenberg?

We analyzed the performance of HATCHet and Battenberg on mirrored events in the prostate dataset. Across the 10 patients, HATCHet2 infers an average of 204 segments per patient where at least one clone has a mirrored-subclonal CNA, present in 18.2% of the tumor genome on average, and overlapping an average of 5.5 TCGA prostate cancer genes. HATCHet can infer mirrored-subclonal CNAs since it uses the same factorization as HATCHet2, but because the mBAF does not respect the invariance of haplotype across samples, it is much less

informative than the mhBAF used in HATCHet2. As a result, HATCHet identifies mirrored events in only 5.5% of the segments in which HATCHet2 identifies a mirrored event. Battenberg is incapable of inferring mirrored-subclonal CNAs, as it will never return a copy-number state (A, B) where $B > A$. We added a section “Mirrored subclonal CNAs identified in prostate cancer dataset” to the Supplement which describes this result for the reader.

Unfortunately, the MASCoTE simulations we used in the initial submission did not contain any mirrored events and due to time constraints we were unable to benchmark the methods on new simulated data with mirrored-subclonal copy number events.

3. How accurate is HATCHet2 when it comes to inferring the number of clones and their proportions on the simulated datasets?

In the main text, we show using pseudo-bulk single-cell WGS data that HATCHet2 is able to recover the two largest clones and their copy-number profiles and mixture proportions. We reproduce the relevant text from Results below.

“Finally, the proportions of these two clones in the four sections is highly concordant between the HATCHet2 pseudobulk analysis and the CHISEL single cell analysis (mean absolute difference 0.040; total variation distance in each sample ≤ 0.0715), providing further evidence to the accuracy of HATCHet2’s results (Fig. 4C).”

To further address this question, we compared HATCHet2 to the other methods using the MASCoTE simulated data and analyzed: (1) the inferred number of tumor clones on multi-sample datasets; and (2) the inferred clone mixture proportions on both multi-sample and single-sample datasets. We reproduce the new supplement text and figure S10 below.

“To evaluate how well HATCHet2 recovers the number and proportion of tumor clones, we evaluated the number and proportions of the clones inferred by HATCHet2, HATCHet, cloneHD, TITAN, and Battenberg on the MASCoTE simulated datasets described in Results. Each method inferred the correct number of tumor clones in about half of instances, except TITAN which systematically underestimated the number of tumor clones (Fig. S10A). We then evaluated the performance of each method in inferring the clone proportions on the subset of datasets where the method correctly inferred the number of clones (as such, TITAN was excluded from this analysis). For all methods, we evaluated each inferred tumor clone against the true tumor clone which most closely matched its proportions. HATCHet2 and HATCHet outperformed the other methods in terms of both mean absolute difference in clone proportions and total variation distance (by 13x and 9.5x, respectively; Fig.S10B-C). HATCHet2 performed similarly to HATCHet on these instances, with a 2.4% higher mean absolute error and a 5% lower total variation distance.

We further evaluated the performance of HATCHet2 against HATCHet treating each sample as an independent dataset ($n=32$) -- for these datasets, both methods were fixed to the correct number of tumor clones, so we evaluated all instances. In this setting, HATCHet2 outperformed HATCHet in both metrics with a 29.9% lower mean absolute difference and 27.3% lower total variation distance (Fig. S10D-E).”

Benchmarking HATCHet2, HATCHet, TITAN, cloneHD, and Battenberg inference of clones and clone proportions. A-C Results from all methods on 8 multi-sample datasets simulated using MASCoTE (Raphael & Zaccaria, 2020). In panels B-C, only those instances for which the corresponding method inferred the correct number of clones are evaluated. For panels B-C, asterisks indicate significance by unpaired t-test: * = $0.01 \leq p < 0.05$, ** = $10^{-3} \leq p < 0.01$, *** = $p < 10^{-3}$. **A** Difference between the true and inferred number of clones from each method on the 8 datasets. **B** Mean absolute difference between the true and inferred clone proportions on multi-sample datasets. **C** Maximum total variation distance between the true and inferred clone proportions on multi-sample datasets. **D-E** Results from HATCHet2 and HATCHet in inferring clone proportions on the same data with each sample run independently ($n=32$ samples), measured by clone inference error (D) and maximum total variation distance (E).

4. Figure 3D how does "clone 1" differ from "clone N" and how does "clone 3" differ from "clone 4"? Are they truly distinct clones with differences in other regions of the genome but not the region of interest for this figure, or is the model inferring multiple similar clones? Including clone profiles similar to Figure 3 B,C for the other clones and highlighting differences between them would be helpful. Similarly, results for other patients from the prostate dataset can be shown in the supplement.

We agree that including clone copy-number profiles would help to clarify the results on the prostate cancer dataset. While this is not as simple as including figures like 3B-C for all patients and clones, since this case in which a sample consists entirely of one clone is uncommon, we added Supplementary Figure S12 which shows the major and minor copy-number profiles and the mean clone proportions for all inferred clones for each prostate cancer patient from the Gundem et al. dataset, including patient A10 which is highlighted in Figure 3. Here we reproduced the clone profiles for patient A10:

In Figure 3D, Clone “N” represents the normal clone as described in the figure legend:

“D Inferred haplotype-specific copy numbers (a,b) (first row) and clone proportions (entries in table) for the normal clone (N) and 4 tumor clones (1-4) for the segment containing the genes ELK4 and SLC45A3.”

5. Additional methodological details are needed, including: details of the E- and M-steps for the mhBAF estimation, does the HMM model include priors, what parameter/hyperparameter settings were used for the analyses throughout?

The E- and M-steps for the mhBAF estimation are identical to those described in the Alleloscope publication. This is described in the Methods section:

“We simultaneously estimate the SNP phasing \hat{h} and the mhBAF \hat{p} by adapting an EM algorithm originally applied to single-cell haplotype frequency estimation (Wu et al., Nature Biotechnol 2021).”

HATCHet2 uses the prior parameters and distributions from the defaults in the HMMlearn Python package `hmmlearn.hmm.GaussianHMM`. (These prior distributions are Dirichlet on the start probabilities [and transition matrix if it is allowed to vary, which is exposed as a parameter], Normal on the cluster means, and inverse gamma or inverse Wishart on the covariance matrix depending on its shape constraints, which are exposed as a parameter.) We added the following text to the Methods section to clarify this:

“Local-global clustering is implemented using the Python class `hmmlearn.hmm.GaussianHMM`. The initial distribution is uniform and the prior distributions and their parameters are kept at their default settings.”

The parameter and hyperparameter settings used to run HATCHet2 (and HATCHet on simulated data as well) is reported in the publication repository as described in the Methods section:

“The HATCHet2 results on both simulated and real data, as well as the HATCHet results on single-sample simulated datasets, are available in the HATCHet2 publication repository: <https://github.com/raphael-group/hatchet2-paper>.”

6. How were user-tunable parameters were selected, and how does modifying them impact performance (minimum number of SNPs/total reads per bin, maximum CN, minimum clone frequency, etc.)?

On simulated datasets, user-tunable parameters were set once for all instances and HATCHet2 automatically selected the number of tumor clones and WGD status. On real datasets, the parameters were manually tuned to optimize results. The detailed parameter settings used for every dataset is included in the publication repository:

<https://github.com/raphael-group/hatchet2-paper>

The documentation for HATCHet2 includes descriptions of all parameters for each step (https://raphael-group.github.io/hatchet/doc_fullpipeline.html#detailed-steps), as well as detailed recommendations on parameter tuning for the steps most sensitive to parameter settings.

4. Binning: increasing the minimum number of SNP/total reads per bin produces larger bins that tend to have a cleaner signal at the cost of fitting small events more poorly, as we describe in the “Analyze different type of data” recommendation page:
https://raphael-group.github.io/hatchet/recommendation_datatype.html
5. Clustering: by default HATCHet2 explores a range of values for the number K of clusters, but users may want to fine-tune the number of clusters. We provide guidance in the “Analyze global clustering” recommendation page:
https://raphael-group.github.io/hatchet/recommendation_clustering.html
 - a. In addition to the plots produced by HATCHet2, a recent tool CNAViz (Lalani et al., *PLOS Comput. Biol.* 2022) can be used to interactively inspect and edit the clustering solution produced by HATCHet2:
<https://github.com/elkebir-group/cnaviz>
6. Inference: by default HATCHet2 automatically selects the clonal cluster, infers WGD status, and chooses the correct number of tumor clones (testing 1-7 tumor clones by default). Users may want to fine-tune these parameters and others, and we provide guidance on doing so in the “Analyze HATCHet’s inference” recommendation page:
https://raphael-group.github.io/hatchet/recommendation_inference.html

The HATCHet2 pipeline automatically produces plots after the clustering step that can be used to troubleshoot binning and clustering, and after the inference step that can be used to troubleshoot the factorization step. Each of these steps can be run and tuned independently or jointly, as the HATCHet2 pipeline is highly modular. The parameters used for each step of HATCHet2, as well as the steps to be run, can be specified a single .ini file (referred to in the documentation as “hatchet.ini”) and run using a single command:

https://github.com/raphael-group/hatchet/blob/master/docs/source/doc_runhatchet.md

7. In Figure 4, it looks like focal amplifications identified by HATCHet2 and CHISEL are quite different. Can the authors explain why this is?

We thank the reviewer for their observation – this impression was driven by a plotting bug that has since been fixed and Figure 4 has been updated. Specifically, to address the reviewer’s comment, we examined chr16 which contained high-copy segments from both methods and found that the segments with high copy number are largely consistent between HATCHet and HATCHet2:

Additionally, after reexamining the HATCHet2 solution on the 10x data more closely, we noticed a haplotype switching bug on chromosome 2 that we were able to fix using a more recent version of the code. While this makes the figure more visually appealing, the clone proportion metrics in the text did not change meaningfully (total variation distance ≤ 0.0715 rather than ≤ 0.072 , and mean absolute error 0.040 rather than 0.039).

Minor comments

1. Naming the modified BAF "mirrored haplotype B-allele frequency" is confusing in that it implies this statistic solely an indicator of mirrored CNAs. The term used in the caption of Figure S6 "minor haplotype BAF" seems (to this reviewer) more appropriate and easily understood.

We thank the reviewer for their comment and have changed all instances of "mirrored haplotype BAF" to "minor haplotype BAF" in the text and figure labels.

2. Similar to the above, calling the HMM "locality aware clustering" in the introduction is unnecessarily confusing. HMMs are common in CNA inference and most readers will likely be familiar with them. The authors do make clear that the simple Gaussian HMM isn't used to infer

integer copy number but only to assign latent states to bins with similar distributions and genomic proximity. That explanation is fine - no need to give it a new term.

We understand the reviewer's concern, but argue that simply stating that we use an HMM would mislead the reader into thinking that HATCHet2's use of an HMM is similar to other CNA inference methods that use HMMs with predefined integer copy numbers and/or global parameters such as purity and ploidy, especially if they do not read the methods or supplement closely.

As a compromise, we changed the term to "local-global clustering" to emphasize that the purpose of this step in HATCHet2 is simply to cluster bins using information about the bins' relative locations, and explicitly call out the use of the HMM in the Introduction when this term is first used.

Second, HATCHet2 improves the identification of clone-specific focal CNAs across multiple samples by...computing a local-global clustering with a Hidden Markov Model (HMM).

We also state more clearly that the local-global clustering is implemented using an HMM in the Methods section:

"We model this clustering problem using a hidden Markov model (HMM) with the following simple structure."

Finally, in the Methods we also describe the differences between our HMM and other uses of HMM's in CNA inference:

HATCHet2 uses a local-global clustering approach based on an HMM that groups bins according to their RDR and mhBAF across all samples as well as their genomic positions.

...

The architecture of this HMM is much simpler than other HMMs used for copy number inference, which have hidden states representing specific copy-number states (e.g. amplification, deletion, LOH, etc) \cite{hmmcopy,titan,remixt} and admixture by normal cells and multiple tumor clones \cite{titan}. The simple HMM used in \ourmethod has fewer parameters to infer. In fact, it has only one additional parameter τ compared to the Gaussian mixture model (GMM) employed by HATCHet \cite{hatchet}; HATCHet's global clustering can be viewed as a "0th-order" HMM with Gaussian emissions.

3. Many CNA callers have a hyperparameter that can be modified to reduce over/under segmentation. How were parameters set for the competing methods?

Parameters for non-HATCHet methods were set as described in Supplementary Note 1 from the HATCHet paper (Zaccaria & Raphael, 2020). Parameters for HATCHet were set to match HATCHet2 for all shared steps. The GMM clustering for HATCHet used "initclusters=100" which

effectively sets the maximum number of clusters to 100 (although the GMM can and often does infer much fewer clusters), and no other hyperparameters related to clustering.

4. Can HATCHet2 detect subclonal WGD or does it assume ploidy is clonal? The supplement indicates that in the simulated datasets any WGD affected all clones.

Yes, HATCHet2 can detect subclonal WGD – it does not assume that all clones share the same ploidy. HATCHet2 detects WGD by testing different values for the most common balanced clonal state (base copy-number state). For a tumor with subclonal WGD, the inferred base copy-number state may be either 1,1 or 2,2 depending on the aneuploidy in the diploid and tetraploid clone genomes, but HATCHet2 is capable of detecting subclonal states (such as subclonal 2,2 or 2,1) regardless of the base copy-number state.

5. For Figure 2, S2, and S3, the RDR and mhBAF values appear identical for all tools. Is this because all of the tools were run with the same input as HATCHet2, or were the inferred CN calls for those tools simply mapped onto the HATCHet2 RDR and mhBAF values for visualization? This should be made clear.

In Figures 2, S2, and S3, each point is a SNP and is placed at the location corresponding to the simple BAF at the locus (variant reads / total reads) or the RDR in the neighboring region.

These quantities do not vary by method. We have added text to the figure legend to clarify this:

“Each point is a small genomic region that contains exactly one SNP with indicated read-depth ratio (RDR) and B-allele frequency (BAF), and is colored by the assigned copy-number state for the corresponding method.”

6. Figure 2 C,D,E,F, Figure 3 B,C, Figure S2 and S3 need a colour legend.

We added supplementary figure S7 which includes the inferred allele-specific copy numbers for each method and clone for the regions shown in Figure 2, Figure S2 (now Figure S3), and Figure S3 (now Figure S4). We added the following text to the caption of Figure 2 to clarify this:

“Full copy-number state legends for panels C-F are reported in Fig. S5.”

In Figure 3, the states in panels B and C are colored by the haplotype-specific copy numbers labeled in Figure 3B. We added the following text to the caption to clarify this:

“B ... Each point is colored by the copy-number state assigned to the bin... C ...Points are colored by the assigned haplotype-specific copy-number state as in (B).”

7. Figure 3 B and C have "fractional copy number" on the y-axis. This isn't defined.

Fractional copy number is directly proportional to the read-depth ratio, but is more closely related to the inferred integer copy numbers. We have added text to the Figure 3 legend to clarify this: "...and fractional copy number (FCN, a rescaling of the read-depth ratio, y-axis)"

8. Typo on page 14 "individual cells".

Thank you for your careful reading. We have corrected this typo.

Second round of review

Reviewer 1

I want to thank the authors for satisfactorily responding to my questions. I'm afraid I still have to disagree with the statement, "To demonstrate the advantages of HATCHet2 and their impact, we focus in detail on two examples involving genes with a key role in prostate cancer: TP53 and CANT1.", when there is, at best, weak evidence that CANT1 has a role in prostate cancer. However, I do not believe that this alone should delay the publication of this manuscript and would leave it up to the authors to decide if they want to revise this statement so it more accurately reflects the presumed role of CANT1.

Reviewer 3

Thank you for the detailed response and new figures, which addressed most of my comments.

Major comments:

1. Thank you for the new benchmarking analysis on focal amplifications. Please reference this and the other new supplementary figures somewhere in the main text so readers can find them.
2. From the authors' response "Across the 10 patients, HATCHet2 infers an average of 204 segments per patient where at least one clone has a mirrored-subclonal CNA, present in 18.2% of the tumor genome on average... HATCHet identifies mirrored events in only 5.5% of the segments in which HATCHet2 identifies a mirrored event." This is a substantial difference and the average number of detected events per patient seems quite high. Given that introduction of the new statistic mhBAF and detection of mirrored-subclonal events is one of the two central claims of this manuscript, I think it is reasonable to: 1) expect benchmarking on simulated data, and 2) state the number of events detected and the difference between HATCHet2 and HATCHet on the prostate dataset in the main text, not the supplement.

Minor comments:

1. Page 2, line 37 still says mirrored-haplotype rather than minor-haplotype.
2. The new track included in the response to reviewers for Figure 2A doesn't appear in the main text.

We thank the reviewers for the constructive feedback on our manuscript. Please see below our point-by-point responses (in blue text) to the comments of each reviewer (black text).

Reviewer #1: I want to thank the authors for satisfactorily responding to my questions. I'm afraid I still have to disagree with the statement, "To demonstrate the advantages of HATCHet2 and their impact, we focus in detail on two examples involving genes with a key role in prostate cancer: TP53 and CANT1.", when there is, at best, weak evidence that CANT1 has a role in prostate cancer. However, I do not believe that this alone should delay the publication of this manuscript and would leave it up to the authors to decide if they want to revise this statement so it more accurately reflects the presumed role of CANT1.

We have modified the text referring to CANT1 as follows below, and we added supporting references:.

"...we focus in detail on two examples involving genes with previously reported roles in prostate cancer: *TP53* [1-2] and *CANT1* [3-4]."

[1] Huang, Hang, et al. "Significance of TP53 and immune-related genes to prostate cancer." *Translational Andrology and Urology* 10.4 (2021): 1754.

[2] Zhang, Wensheng, et al. "Deciphering the increased prevalence of TP53 mutations in metastatic prostate cancer." *Cancer Informatics* 21 (2022): 11769351221087046.

[3] Gerhardt, Josefine, et al. "The androgen-regulated Calcium-Activated Nucleotidase 1 (CANT1) is commonly overexpressed in prostate cancer and is tumor-biologically relevant in vitro." *The American journal of pathology* 178.4 (2011): 1847-1860.

[4] Yang, Wei, Zhidong Liu, and Ting Liu. "Pan-cancer analysis predicts CANT1 as a potential prognostic, immunologic biomarker." *Cellular Signalling* (2024): 111107.

Reviewer #3: Thank you for the detailed response and new figures, which addressed most of my comments.

Major comments:

1. Thank you for the new benchmarking analysis on focal amplifications. Please reference this and the other new supplementary figures somewhere in the main text so readers can find them.

We have updated the main text to refer at least once to each of the supplemental figures and/or the supplemental results sections referencing them.

2. From the authors' response "Across the 10 patients, HATCHet2 infers an average of 204 segments per patient where at least one clone has a mirrored-subclonal CNA, present in 18.2% of the tumor genome on average... HATCHet identifies mirrored events in only 5.5% of the segments in which HATCHet2 identifies a mirrored event." This is a substantial difference and

the average number of detected events per patient seems quite high. Given that introduction of the new statistic mhBAF and detection of mirrored-subclonal events is one of the two central claims of this manuscript, I think it is reasonable to: 1) expect benchmarking on simulated data, and 2) state the number of events detected and the difference between HATCHet2 and HATCHet on the prostate dataset in the main text, not the supplement.

To measure the performance of HATCHet2 in recovering mirrored-subclonal CNAs, we generated a new dedicated set of simulated datasets, which we describe in Section S8-S9 of the supplement and reference in the main text. Based on these 150 simulated datasets, we show that HATCHet2 has substantially higher performance in identifying mirrored-subclonal CNAs, although interestingly HATCHet is able to identify some mirrored-subclonal events in 10/150 of the datasets. We reproduce the supplementary results Section S9 and Figure S11 describing these new benchmarking results below. In addition, we moved the text describing mirrored-subclonal CNAs inferred in the Gundem et al. prostate cancer dataset from the supplement to the results section of the main text.

To evaluate the performance of HATCHet2 in recovering mirrored-subclonal CNAs, we generated a total of 150 simulated datasets as described in Section S8. These simulated datasets include all combinations of 10 simulated genomes (5 diploid, 5 tetraploid) with the 3 sets of clone proportions described above, using 5 different random number generator seeds for each combination. We applied HATCHet2 and HATCHet to these simulated datasets and measured the AASAPGP and accuracy, both on the whole genomes and restricted to mirrored-subclonal events (i.e., measuring performance on only those states (A, B) where B>A) (Fig. S11). We refer to these restricted metrics as “mirrored accuracy” and “mirrored AASAPGP”, respectively. HATCHet2 outperformed HATCHet in all of these metrics ($p < 10^{-14}$, average improvement $\geq 30.6\%$) and especially in terms of mirrored-subclonal states ($p < 10^{-62}$, average improvement $\geq 1100\%$). HATCHet performs especially poorly on 3 of the tetraploid genomes, with an average AASAPGP of 0.36 and average accuracy of 0.13, compared to 0.92 and 0.86, respectively, for HATCHet2. Indeed, HATCHet failed to identify any mirrored-subclonal CNAs in 140/150 instances, with the exceptions being diploid genomes 1 and 2 with clone proportions 1.

To further analyze the difference in performance on mirrored-subclonal CNAs, we examined the HATCHet2 and HATCHet results from two simulated datasets in detail: one dataset in which HATCHet was able to identify some mirrored-subclonal CNAs (with diploid genome 1, sample clone proportions 1, and random seed 0; Fig. S11E-J), and one in which it was not (with the same diploid genome 1, sample clone proportions 2, and random seed 0; Fig. S11K-N). In the first dataset, the two clusters containing mirrored-subclonal CNAs (denoted by red circles) have mhBAF below 0.5 in the first sample (Fig. S11E,H), so the mBAF used by HATCHet is equal to the mhBAF used by HATCHet2 for these clusters in this sample and thus HATCHet is able to identify the mirrored-subclonal states (mirrored accuracy 0.992). However, since HATCHet does not retain haplotype information across samples, HATCHet is unable to assign the mirrored-subclonal states correctly to clones and thus incorrectly infers that both of the

other samples contain the same clone (mirrored AASAPGP 0.3; Fig. S11I-J). In contrast, HATCHet2 using the mhBAF correctly assigns the mirrored-subclonal states to distinct clones in the single-clone samples (mirrored accuracy 0.997, mirrored AASAPGP 0.988; Fig. S11F-G). The second dataset consists of the same simulated genome (i.e., the same copy-number states in all segments for both clones) as the first example dataset, but with different sample clone proportions. In this case, HATCHet2 recovers the mirrored-subclonal CNAs in both clones (mirrored accuracy 0.996, mirrored AASAPGP 0.915; Fig. S11K-L), but HATCHet is unable to distinguish between events affecting different haplotypes and thus identifies no mirrored-subclonal CNAs (mirrored accuracy 0, mirrored AASAPGP 0; Fig. S11M-N).

Figure S11: Results of HATCHet2 and HATCHet on simulated data with mirrored-subclonal CNAs. A-B Results from both methods on all copy-number states, measured by AASAPGP (A) and accuracy (B). C-D Results from both methods on mirrored-subclonal states only (i.e., only those states (A, B) where B > A), measured by AASAPGP (C) and accuracy (D). E-J Results from HATCHet2 and HATCHet on an example simulated dataset in which HATCHet achieves high accuracy on mirrored states (diploid genome 0, sample clone proportions 1, random seed 0). Panels E-G show the HATCHet2 results, and H-J show the HATCHet results. The title of each panel indicates the ground truth simulated clone proportions. Each point is a copy-number bin, colored by its assigned states across the inferred tumor clones. The labeled points with black outlines indicate the expected position of the assigned

copy-number states given the inferred clone proportions. Inferred mixture proportions for each sample are shown in the legend. Red circles highlight some mirrored-subclonal copy-number states. **K-N** Results from HATCHet2 and HATCHet on an example simulated dataset in which HATCHet was not able to identify any mirrored-subclonal states (diploid genome 0, sample clone proportions 2, random seed 0). Panels K-L show the HATCHet2 results, and M-N show the HATCHet results.

Minor comments:

1. Page 2, line 37 still says mirrored-haplotype rather than minor-haplotype.

Thank you for your attention to detail. We have corrected these errors.

2. The new track included in the response to reviewers for Figure 2A doesn't appear in the main text.

Thank you for your attention to detail. We have corrected these errors.